# Assessing the dynamics of vegetation productivity in circumpolar regions with different satellite indicators of greenness and photosynthesis

Sophia Walther[1], Luis Guanter[1], Birgit Heim[2], Martin Jung[3], Gregory Duveiller[4], Aleksandra Wolanin[1], and Torsten Sachs[1]

[1]Helmholtz Centre Potsdam - GFZ German Research Centre for Geosciences
[2]Alfred-Wegener-Institut - Helmholtz-Zentrum für Polar- und Meeresforschung Potsdam
[3]Max Planck Institute for Biogeochemistry, Jena
[4]European Commission, Joint Research Centre, Directorate D – Sustainable Resources - Bio-Economy Unit, Ispra, Italy

*Correspondence to:* Sophia Walther, sophia.walther@gfz-potsdam.de (now at Max-Planck-Institute for Biogeochemistry, Jena, Germany, swalth@bgc-jena.mpg.de)

**Abstract.** High latitude treeless ecosystems represent spatially highly heterogeneous landscapes with small net carbon fluxes and a short growing season. Reliable observations and process understanding are critical for projections of the carbon balance of climate sensitive tundra. Spaceborne remote sensing is the only tool to obtain spatially continuous and temporally resolved information on vegetation greenness and activity in remote circumpolar areas. However, confounding effects from persistent clouds, low sun elevation angles, numerous lakes, widespread surface inundation, and the sparseness of the vegetation render it highly challenging. Here, we conduct an extensive analysis of the timing of peak vegetation productivity as shown by satellite observations of complementary indicators of plant greenness and photosynthesis. We choose to focus on productivity during the peak of the growing season as it importantly affects the total annual carbon uptake. The suite of indicators are: (1) MODIS-based vegetation indices (VIs) as proxies of the fraction of absorbed photosynthetically radiation; (2) APAR estimated as VIs combined with estimates of absorbed photosynthetically active radiation; (3) sun-induced chlorophyll fluorescence (SIF) serving as a proxy for photosynthesis; (4) vegetation optical depth (VOD), indicative of total water content; and (5) empirically upscaled modelled gross primary productivity (GPP). Averaged over the pan-Arctic we find a clear order of the annual peak as APAR <= GPP < SIF < VIs / VOD. SIF as an indicator of photosynthesis is maximized around the time of highest annual temperatures. Model GPP peaks at a similar time like APAR. The time lag of the annual peak between APAR and instantaneous SIF fluxes indicates that the SIF data do contain information on light-use efficiency of tundra vegetation, but further detailed studies are necessary to verify this. Delayed peak greenness compared to peak photosynthesis is consistently found across years and land cover classes. A particularly late peak of the normalized difference vegetation index (NDVI) in regions with very small seasonality in greenness and a high amount of lakes probably originates from artefacts. Given the very short growing season in circumpolar areas, the average time difference in maximum annual photosynthetic activity and greenness/growth of 3 to 25 days (depending on the data sets chosen) is important and needs to be considered when using satellite observations as drivers in vegetation models.

# 1 Introduction

Landscapes in circumpolar regions are characterized by sparse vegetation, bare soil, rocks, large surface areas inundated by open water and a long snow covered period. Despite large carbon amounts being stored in the often permanently frozen grounds, net fluxes of carbon between the land surface and the atmosphere are small and their $CO_2$ balance is close to neu-
trality (McGuire et al., 2012). Because of their strong sensitivity to environmental conditions, carbon exchange processes are highly variable in space and time (Olivas et al., 2011; Pirk et al., 2017; Lafleur and Humphreys, 2008; Welker et al., 2004) and an ecosystem might switch between being a carbon sink or source from year to year depending on the weather conditions (Huemmrich et al., 2010b).

Warming happens at accelerated rates compared to middle and lower latitudes (AMAP, 2012). The carbon budgets of both the tundra ecosystem and the Arctic boreal zone as a whole are undergoing major changes – with possibly strong positive feedbacks to climate (Pearson et al., 2013). The future evolution of net ecosystem exchange (NEE) and its component fluxes gross primary productivity (GPP) and respiration in Arctic landscapes is highly uncertain. Higher temperatures, the accompanying mineralization as well as higher atmospheric $CO_2$ concentrations fertilize vegetation (Yi et al., 2014; Zhu et al., 2016; Welker et al., 2004). Accordingly, changes in species composition (Chapin et al., 1995) are observed and satellite records indicate a greening in large regions in the Arctic (Jia et al., 2003; Verbyla, 2008). This is interpreted as increased growth (Racine et al. in Stow et al., 2004; Elmendorf et al., 2012; Huemmrich et al., 2010a; Chapin et al., 1995) or even woody encroachment into the tundra (Racine et al. in Stow et al., 2004; Dass et al., 2016; Sturm et al., 2001). Yet, higher leaf mass and growth do not in every case necessarily linearly translate into enhanced GPP as increased growth might also cause enhanced self-shading and lower nitrogen amounts per unit leaf area (Street et al., 2007; McFadden et al., 2003). A warmer climate might extend the snow free period (Myneni et al., 1997) but there are contradicting indications of whether (Ueyama et al., 2013b; Lund et al., 2010; Kross et al., 2014) or not (Gamon et al., 2013; Oberbauer et al., 1998; López-Blanco et al., 2017; Lafleur and Humphreys, 2008) a longer growing season enhances seasonal carbon uptake and growth. Photosynthetic activity and plant growth further depend on soil moisture conditions (Gamon et al., 2013; Opała-Owczarek et al., 2018; Lafleur and Humphreys, 2008; Welker et al., 2004) and therefore, warming and shorter and shallower snow packs do not necessarily increase productivity (Zhang et al., 2008; Gamon et al., 2013; Yi et al., 2014; Huemmrich et al., 2010b, a; Parida and Buermann, 2014). Soil warming promotes thaw and stronger drainage. Heterogeneous respiration and carbon emissions to the atmosphere are stimulated in warmer soils at lowered water table depth (Billings et al., 1982; Yi et al., 2014; Oechel et al., 1993; Huemmrich et al., 2010b; Commane et al., 2017). The balance between photosynthetic carbon uptake and respirational losses is further modulated by permafrost disturbances (Cassidy et al., 2016). Polar treeless regions are spatially highly heterogeneous ecosystems (Welker et al., 2004) but with widespread full vegetation cover. NEE, GPP and respiration are governed by variable conditions regarding wetness and temperature, micro-topography, geomorphology and type and acidity of the soils (Kwon et al., 2006; Walker et al., 1998; Olivas et al., 2011; Emmerton et al., 2016; Pirk et al., 2017). It is not clear whether, where and when the land surface in Arctic tundra actually acts as a sink or source of $CO_2$ (Cahoon et al., 2012; McGuire et al., 2012) and what the direction and magnitude of changes in altered climatic conditions will be (Oechel et al., 1993; Billings, 1987; Sitch et al.,

2007). This has given rise to extensive and long-term project studies of the Arctic like the Arctic-Boreal Vulnerability Experiment (ABoVE, https://above.nasa.gov/about.html) or the Carbon in Arctic Reservoirs Vulnerability Experiment (CARVE, https://carve.jpl.nasa.gov/Missionoverview/), both of which are not limited to $CO_2$).

Observing carbon fluxes in these inaccessible and remote areas is difficult. Several long-term monitoring sites exist where phenological observations, spectral reflectance as well as gas flux measurements are conducted *in-situ*, both under natural conditions and in manipulative experiments. Many studies can be found in the literature that evaluate eddy-covariance or chamber gas flux measurements with respect to spatial patterns of NEE at a fixed point in time, or *in-situ* NEE integrated over the growing season and its variations between years (López-Blanco et al., 2017; Lund et al., 2010; Ueyama et al., 2013a; McFadden

et al., 2003; Williams and Rastetter, 1999; Marushchak et al., 2013; Kross et al., 2014). However, only few sites exist compared to temperate regions and observations are usually not done in a continuous manner over the complete year but during individual measurement campaigns or dedicated periods during the growing season. Even if automated instrumentation can provide more continuous measurements all along the year, it is still hampered by the difficulty of access in case of equipment failure. Compared to more temperate sites, tundra poses additional challenges on the calculation of NEE and its component

fluxes GPP and respiration (Pirk et al., 2017). Due to the small magnitudes of the net fluxes, different flux calculation methods might even differ in whether they indicate a source or a sink at a given time (Pirk et al., 2017). Snow and soil freezing can act as a barrier for gas exchange with the atmosphere and cause a temporal decoupling between the registration at the sensors and when the gas concentrations have actually been changed by heterotrophic respiration in the soil (Arneth et al., 2006) or by photosynthesis by evergreens under the snow (Starr and Oberbauer, 2003). Continuous illumination during polar day chal-

lenges temperature-respiration relationships obtained from nighttime data (Runkle et al., 2013). Further, the heterogeneity of the landscape poses limits to the spatial representativeness of the relationships between the carbon fluxes and meteorological and soil conditions that have been identified *in-situ* (Pirk et al., 2017; Tuovinen et al., 2018). Therefore, in spatial up-scaling exercises (Ueyama et al., 2013a; Marushchak et al., 2013; Huemmrich et al., 2013; Tramontana et al., 2016) strong extrapolations are necessary. Yet, the modelling of the future evolution of the vegetation and carbon fluxes (including their timing and

magnitude) in circumpolar areas requires an understanding of the component fluxes GPP and respiration as well as accurate spatially and temporally explicit observations of their drivers.

    Satellite remote sensing can help to constrain the component flux of GPP and additionally to extend point observations to larger areas with repetitive coverage in time. Depending on the monitoring approach, different assets and limitations need to be

considered for inferring GPP. Optical reflectance measurements can give an indication of the abundance of green plant material and hence photosynthetic potential. From spectral observations of greenness, information can be inferred on the fraction of incident photosynthetically active radiation (PAR) that is absorbed (fAPAR) and can potentially be used for carbon fixation. Following the concept of the light-use efficiency of plant productivity by Monteith (1972), the amount of absorbed radiation (APAR, the product of fAPAR and incident PAR) is an important determinant of spatial and seasonal variations in GPP to-

gether with the efficiency with which the absorbed energy is used in carbon fixation. Site-level studies have confirmed a highly

linear relationship between APAR and GPP (Huemmrich et al., 2010a, b). Indeed, in the last decades, spatial extrapolations of *in-situ* observations of carbon fluxes in tundra and peatland showed the skill of indicators of greenness (leaf area index, LAI, or reflectance based indices like the NDVI or the green ratio) as a predictor for GPP and NEE (Ueyama et al., 2013a, b; Chadburn et al., 2017; McFadden et al., 2003; Williams and Rastetter, 1999; Street et al., 2007; Marushchak et al., 2013).

At many sites, mosses make up twice or trice the biomass of vascular plants. However, their photosynthetic capacity is much lower (Yuan et al., 2014; Williams and Rastetter, 1999; Huemmrich et al., 2013; Zona et al., 2011), and their seasonality is often dissimilar (Gamon et al., 2013) as a consequence of their different sensitivities to environmental conditions (Zona et al., 2011). Micro-topography affects moisture conditions, even within small elevation changes of about one meter (Olivas et al., 2011; Gamon et al., 2013; Pirk et al., 2017). As a consequence, distinct spatial distributions of the plant functional types and

highly variable patterns of photosynthetic light-use efficiency are observed. Vascular plants prefer lower, wetter places and their growth increases biomass, productivity, NDVI, and LAI. However, when the ground becomes drier, NDVI will increase, but actual productivity decline (Olivas et al., 2011; Gamon et al., 2013; Buchhorn et al., 2013). Consequently, the wetness of the surface confounds the interpretation of spectral reflectance with respect to productivity, which is problematic as soils are often water-saturated (Stow et al., 2004). Next to the confounding effect of moisture on spectral reflectance, changes in GPP

have been observed to not necessarily translate into changes in NDVI (Olivas et al., 2011). The dynamics of field measurements of NDVI are not necessarily related to total photosynthetic phytomass, but rather driven by only parts of the live foliar component (Riedel et al., 2005). In addition to these challenges, spectral reflectance observations are affected by large signals from the background and shadows cast by microtopography and vegetation itself. Snow and open water from the many rivers, small ponds and thaw lakes (globally, the highest abundance and areal coverage of lakes is between 55 and 75° N, Verpoorter

et al., 2014) as well as litter and dry plant material influence the spectra with seasonally changing extents. Further, persistent cloud cover, low illumination and viewing geometry (Stow et al., 2004; Laidler and Treitz, 2003) and the relatively large pixel size compared to the high heterogeneity of the landscape render reflectance-based observations of circumpolar productivity difficult.

Recently, independent and complementary approaches to spectral reflectance have become available to remotely study veg-

etation dynamics. First, sun-induced chlorophyll fluorescence (SIF) is an electromagnetic signal emitted by chlorophyll as a 'by-product' of photosynthesis. Because it is directly related to photosynthetic activity (e.g. Porcar-Castell et al., 2014) it is expected to give a more direct and accurate picture of actual photosynthesis (as compared to greenness/ growth) and is much less affected than vegetation indices by open water, snow or background effects, the heterogeneity of the land surface and plant functional types. However, the footprints of the sensors from which SIF measurements are available for several years are very

large and integrate over many different growing conditions. Further, the SIF signal is generally weak in tundra regions due to the low vegetation abundance and photosynthetic rates and in combination with low illumination angles subject to high noise levels.

A second type of complementary satellite information lies in passive microwave remote sensing. Specifically, vegetation optical depth (VOD) is a radiometric variable describing the attenuation of microwave radiation emitted from the soil and the

vegetation itself due to the water contained in the canopy. It can therefore be directly related to vegetation water content and

biomass. VOD increases with vegetation density, but is strongly controlled by vegetation emission in very dense vegetation (Liu et al., 2011). Depending on the wavelength of observation, the signal is sensitive to different depths in the canopy and objects of variable sizes (e.g. small objects like leaves versus large trunks or branches). Following Teubner et al. (2018), in moderately and sparsely vegetated areas, there is a chain of proportionalities from VOD to GPP. VOD indicates total water content, which is related to leaf area, which in turn is an important determinant of GPP. In their comprehensive study, Teubner et al. (2018) evaluated the temporal behaviour between different VOD data sets, model GPP and SIF, and found widespread high positive correlations both between the raw time series as well as patterns of anomalies globally. Although dynamics in tundra vegetation have not been explored explicitly, correlations between VOD and GPP were consistently high in landscapes characterized by shrubs, grasses or sparse vegetation. Similarly, highest correlations between phenological dates derived from VOD and vegetation indices were obtained in low biomass regions (Jones et al., 2011). VOD observations are insensitive to cloud cover and to variations in day light, a strong advantage in the high latitudes of interest in our study. However, as for SIF, currently available satellite observations have a coarse spatial resolution compared to optical measurements. Further, careful corrections of effects of soil moisture, open water and frozen grounds, snow and ice, amongst others are necessary in the retrieval, and it is therefore not clear whether VOD can be a useful parameter to evaluate vegetation dynamics in the specific context of tundra.

Neither greenness nor SIF nor VOD can directly be translated into the amount of carbon taken up through photosynthesis. Nevertheless, they all represent important observation-based driving variables for the modelling of tundra carbon exchanges at landscape scale and over multiple years (e.g. Luus et al., 2017). Therefore, their ability to accurately represent the timing and relative changes of photosynthetic activity and growth is of key importance for realistic model estimates of carbon fluxes. In this study, we compare the timing of the peak growing season as indicated by several satellite vegetation indices, VOD and SIF in circumpolar treeless regions. We aim at analysing their complementary information content with respect to maximum greenness and photosynthetic activity - despite all above-mentioned challenges - and relate them to environmental conditions. In addition, GPP empirically up-scaled from eddy-covariance observations using satellite measurements of different variables describing the land surface and meteorological reanalysis data (Tramontana et al., 2016) is included in the study. In doing so, a comprehensive evaluation of several state-of-the-art satellite-based products is achieved in this study with a special focus on the timing of the peak growing season, as this represents the most important period with respect to total annual carbon uptake. In addition to the use of the broad array of complementary spaceborne data sets, we perform this analysis for the total circumpolar pan-Arctic treeless regions and it therefore represents an extension with respect to the majority of published tundra ecosystem studies that are mostly confined to specific regions, like Alaska.

## 2   Methods and material

### 2.1   Methods

The different vegetation proxies are evaluated at 0.5° spatial resolution and with daily sampling. A temporal running mean in a window of 16 days is applied to all data sets. The resulting data still contain values for every day in a year, but the effective temporal resolution corresponds to 16 days. Gaps due to missing data are not aligned between data sets. The timing of the annual maximum is defined as the average day of year (DOY) of all days at which the values exceed the 95[th] quantile of all valid values of the time series in a year in a given pixel. Because of frequently low data quality and long and intermittent data gaps in those high latitude regions of interest, we mostly base our analysis on multi-year averages of the DOYs of annual maximum (henceforth avg.peak). We test the uncertainties by bootstrapping each annual time series 50 times per year and pixel for selected vegetation proxies (DOYs sampled consistently between vegetation proxies, with replacement, and restricted to DOYs 100-300).

We use tower eddy covariance measurements as a test of consistency between satellite observations sampled at the towers and site-level conditions.

### 2.2   Data sets

#### 2.2.1   Environmental variables

Air temperature at two meters height (t2m) every six hours between 2007–2015 is obtained from ERAInterim reanalysis data (Dee et al., 2011) and aggregated to 16-day temporal resolution with daily sampling.

Daily global radiation (Rg) for the years 2007-2015 is obtained from measurements of the Clouds and the Earth's Radiant Energy System (CERES Ed4A, Wielicki et al., 1996; Doelling et al., 2013) onboard the Aqua and Terra satellites. From the 1° spatial resolution product (the 'SYN1deg-Day product', all-sky surface shortwave downward fluxes, initial fluxes) we disaggregate to 0.5° spatial resolution by bilinear interpolation. Subsequently, daily data are averaged in a daily moving window of 16 days.

We further include surface soil moisture (SM, 0-10cm depth in $m^3/m^3$) model results from the GLEAM project (v3.1a, Miralles et al., 2011; Martens et al., 2017). GLEAM is provided at daily temporal and 0.25° spatial resolution for the years 2007–2015. For the analysis we aggregate them to 0.5° and 16-day resolution with daily sampling. In case of moisture-related variables we will explore the timing of the annual minimum as well in order to get an indication of potential moisture stress or confounding effects on reflectance measurements. Accordingly, the timing of minima are defined as the average of all DOYs at which the values are below the 5[th] quantile of all valid values in a year in a given pixel.

#### 2.2.2   Reflectance-based indices

We use MODIS reflectance measurements to obtain the enhanced vegetation index EVI (Huete et al., 2002), the normalized difference vegetation index NDVI (Tucker, 1979), and the near infra-red reflectance of vegetation NIRv (Badgley et al., 2017)

as proxies of greenness for the years 2007–2015. These indices have been calculated from Nadir Bidirectional reflectance distribution function Adjusted Reflectance (NBAR) from the MODIS MCD43C4v006 product (MCD43C4: NASA LP DAAC and Science , EROS) at 0.05°. This means that the reflectance values are modelled to a value as if viewed from directly above. After quality check (only pixels with bi-directional Reflectance Distribution Function (BRDF) Quality flags 0, 1 retained) and snow filter (all pixel values containing any snow removed) the data are aggregated to 0.5° spatial resolution and left at their native temporal resolution of 16 days with daily sampling. We will refer to these throughout the manuscript as EVI, NDVI and NIRv.

As the amount of incoming photosynthetically active radiation is proportional to the total downwelling shortwave radiation we calculate an estimate of APAR as the product of global radiation and EVI (denoted EVI.Rg) or NDVI (denoted NDVI.Rg), both of which are here assumed to be a valid approximation of fAPAR. In the following we will refer to both of them together as APAR, and otherwise separate between EVI.Rg and NDVI.Rg.

We additionally include the MODIS vegetation index products of NDVI and EVI from Aqua MYD13C1v006 (MYD13C1: NASA LP DAAC and Science , EROS) and Terra MOD13C1v006 (MOD13C1: NASA LP DAAC and Science , EROS) in the analysis. In contrast to EVI, NDVI and NIRv from the MCD43C4 data, those are obtained from reflectances with different viewing angles that do not necessarily correspond to nadir. Including them in the comparison can therefore help to get an idea of the influence of directional effects on the seasonality and of the consistency of the results. From the 0.05° products generated with an 8-day frequency using a period encompassing the last eight and the following eight days of acquisitions, we removed data that do not have good quality using the VI quality indicator. We further remove pixel values that are flagged as cloudy, containing snow or ice or those that were not processed as indicated by the quality reliability flag. The remaining pixel values are aggregated to 0.5° grid cells. Throughout the manuscript we will name these EVI.VIproduct and NDVI.VIproduct or refer to both of them together as MODIS VIproduct. The data from the MODIS VIproduct are different from all other datasets in that they are sampled every eight days, not daily.

### 2.2.3   Vegetation optical depth and land surface parameters

A data set of various land parameters simultaneously derived from passive microwave measurements of the AMSR-E onboard Aqua and AMSR2 onboard GCOM-W1 is used for the years 2007-2015 (v2, here called AMSR-E/2,  Du et al., 2017b, a). The data records are combined but not continuous. In 2011 data are available until DOY277 and restarting after that in 2012 only at DOY206. Because the peak growing season is covered in 2011, we do use data from 2011, but not from 2012. Of the observations made in descending orbits with an equatorial crossing at 1:30 AM we use VOD and volumetric surface soil moisture (0-1 cm depth in $cm^3/cm^3$) derived from X-band (10.7 GHz) as well as estimates of the fraction of open water. We use the descending orbit as retrievals are generally more accurate when vertical temperature gradients are low (Liu et al., 2011). The retrieval specifically accounts for the effects of open water on VOD and surface soil moisture (Du et al., 2017b). The accompanying quality flags are used to remove all pixel values observed under non-favourable conditions with respect to frozen soils, snow, ice or large areas of open water on the surface, very dense vegetation, precipitation, radio frequency

interference or microwave signal saturation. The daily files with the native 25 km resolution data in an EASE-grid projection are first quality filtered, then reprojected to 0.25° longitude/latitude relative to WGS84 and subsequently aggregated to 0.5°. For temporal consistency we aggregate to 16 days with daily sampling as in all other data sets.

### 2.2.4 Sun-induced chlorophyll fluorescence

Sun-induced chlorophyll fluorescence (SIF) as a proxy of photosynthetic activity is retrieved from GOME-2 measurements onboard Metop-A at 740 nm (Köhler et al., 2015, ftp://ftp.gfz-potsdam.de/home/mefe/GlobFluo/GOME-2/, it will henceforth be called SIF GFZ) for January 2007 until the end of 2015. We remove the individual measurements that have unfavourable observational conditions, namely those that have an effective cloud fraction of more than 50% (based on the FRESCO cloud mask that is provided with GOME2 L1b data which is based on reflectances in the O2-A band, Wang et al., 2008), those that are measured before 8 o'clock or after 14 o'clock local solar time (which is important as in high latitudes during solar day additional measurements in the evening are possible but subject to high noise) or under sun-zenith angles of more than 70°, and those whose retrieval resulted a residual sum of squares larger than 2 W m$^{-2}$ sr$^{-1}$ $\mu$m$^{-1}$. We also remove those SIF values for which no cloud fraction information is available. The individual remaining measurements are aggregated to 0.5° resolution based on the centre coordinates of a given footprint over the 16-day intervals like in the MODIS data for each individual year to obtain a time series.

We added to the comparison SIF data retrieved from GOME-2 with a slightly different method (Joiner et al., 2013, 2016, https://avdc.gsfc.nasa.gov/pub/data/satellite/MetOp/GOME_F/, V26, henceforth SIF NASA). The individual measurements are filtered in the same way as for the SIF GFZ data set, except for the fact that the data are delivered filtered for an effective cloud fraction of smaller than 0.3 (we treat negative cloud fractions as zero). The SIF NASA uses a similar, but not identical method to obtain effective cloud fractions by using cloud radiance fractions at 865nm (Joiner et al., 2012). We then average in the same way spatially and temporally as before for the years 2007–2015.

As SIF represents an instantaneous observation at a given time of the day and a comparison to GPP seasonality would be hampered by the fact that GPP represents an average daily value (Zhang et al., 2018), additional comparisons are carried out to the SIF observations scaled to daily values (henceforth SIF.daily.int GFZ). By a geometrical approximation of incoming PAR by the cosine of the sun-zenith angle, the correction to daily values is achieved by multiplication of the instantaneous SIF with the ratio of the daily integrated (in 10-min steps) cosine of the sun-zenith angle and the cosine of the sun angle at the time of measurement. This correction is expected to account for the effects of seasonally and daily changing illumination. Caution is warranted for this correction, as it assumes that the same environmental conditions prevail over the entire day and it may further amplify noise.

SIF can be approximated in a similar way like GPP following Monteith (1972) as the product of fAPAR, PAR (approximated as cos(SZA)) and the efficiency with which the energy is used in fluorescence emission. Hence, also the comparison of SIF to vegetation indices is more appropriate if one accounts for the illumination effects on SIF that are not included in the greenness indices. In that case, the SIF values are normalized by the cosine of the sun-zenith angle at the time of measurement and used

in the analysis (henceforth SIF.cosSZA GFZ and SIF.cosSZA NASA). According to the SIF-Monteith model, SIF.cos(SZA) therefore represents a convolution of canopy fAPAR and the efficiency of fluorescence emission.

As a cross check of the plausibility of the GOME-2 SIF additional comparisons to SIF at 757nm retrieved from OCO-2 are done using the OCO-2 SIF lite files (B8100r) from September 2014 to mid October 2017 (OCO-2 Science Team/Michael Gun-
son, 2017; Frankenberg et al., 2014; Sun et al., 2018). We filter all measurements taken with a sun-zenith angle of less than 70°, in nadir mode and over regions whose IGBP land cover is not water, forest, crops, urban or mosaic. Samplings of OCO-2 SIF (henceforth OCO2) and OCO-2 SIF corrected for illumination conditions by division by cos(SZA) in the same way as for the GOME-2 measurements (henceforth OCO2.cosSZA, OCO2 and GOME-2 have different overpass times and therefore different instantaneous illumination at the time of measurement) are averaged to a climatology based on 16-day averages sampled daily
as a spatial average over different smaller regions of interest. The regional averaging is necessary as OCO-2 has no continuous sampling like GOME-2.

### 2.2.5  FLUXCOM model GPP

Another indicator of photosynthetic activity is provided by the GPP model simulations from the FLUXCOM initiative
(http://www.fluxcom.org/products.html,  Tramontana et al., 2016). Relationships between land surface and environmental variables and land-atmosphere energy and carbon fluxes learned at FLUXNET eddy-covariance sites in the La-Thuile data set (http://fluxnet.fluxdata.org/data/la-thuile-dataset/) are spatially up-scaled to the globe using a set of machine-learning techniques. FLUXCOM GPP is generated in two set-ups, the 'remote sensing set-up (RSGPP)' and the 'meteorology + remote sensing (METGPP)' set-up. The former one uses satellite-observed land surface conditions to estimate GPP at 8-daily tem-
poral resolution and 1/12° and we use the years 2007–2015. The METGPP represents an ensemble of GPP where the mean annual cycle of land surface conditions and additional information on actual meteorological conditions from reanalysis is used in the prediction at 0.5° and daily resolution. We restrict the METGPP data to the years 2007–2010 as for those years simulation results from all ensemble members are available. We aggregate to 16-day averages sampled every day, consistent with the MODIS sampling in the MCD43C4v006 data. The RSGPP is linearly interpolated to daily values at 1/12°. Subsequent ag-
gregation to 0.5° and running means over 16 days match the spatio-temporal resolution to all other data sets. Together RSGPP and METGPP are referred to as model GPP.

### 2.2.6  Land cover

We use the ESA CCI land cover classification and aggregate it to broader classes of moss, bare/sparse, grass/herbaceous,
woody, water and other. The ESA CCI provides a tool to convert discrete land cover classes to continuous vegetation fractions (http://maps.elie.ucl.ac.be/CCI/viewer/download/ESACCI-LC-Ph2-PUGv2_2.0.pdf and  Poulter et al., 2015) and we use it to obtain land cover fractions from the native 300 m pixels in 0.5° pixels for the period 2008-2012 (Fig. A1). The distribution of mosses in these products is expected to be problematic because it is a complicated class to characterise for global land cover

products. Being partly based on regional maps with varying thematic detail in their legends, it is possible that this moss class in the ESA CCI is not always accurately identified over certain regions, explaining why moss cover is barely indicated in Siberia.

### 2.2.7 Eddy-covariance derived GPP

We use eddy-covariance measurements from a subset of the FLUXNET2015 data (http://fluxnet.fluxdata.org/, please note that
FLUXCOM has been trained on the earlier La Thuile dataset). We chose sites based on data availability after 2007, situated north of 55° and not characterized as forest (details on sites and site-years used are provided in table A1). We use daily GPP derived using nighttime (NT) and daytime (DT) partitioning and aggregate it to 16day-averages masking out those days where less than 80% of good measured and gap-filled NEE values were available (with variable Ustar threshold and reference selected on the basis of the model efficiency). Site-level air temperature (consolidated by ERA-Interim and the MDS algorithm) has
been filtered for good quality and gap-filling in the same way.

We also include daily GPP derived from NEE by night-time partitioning (Reichstein et al., 2005) from measurements in Cherski/Russia (control site RU-Ch2, data courtesy Mathias Göckede, Max-Planck-Institute for Biogeochemistry, Jena, Germany) where only days are used in the average with short data gaps filled (less than one day).

## 2.3   The study area

Operational satellite-derived land cover data sets exhibit substantial differences in the classes they assign to circumpolar regions. We compared the ESA CCI land cover, GlobeLand30 by Chen et al. (2015) and the IGBP classification from MODIS MCD12C1 and found that classification of the same area can range from barren to grasslands to open shrub lands, depending on the chosen dataset (not shown). A clear and generally accepted delineation of a class 'tundra' is not given. We therefore
define our study area based on tree cover as 'polar treeless regions'.

Global data on annual forest cover gain and loss have been provided by Hansen et al. (2013) based on Landsat images. We take 2009 as representative for the period of investigation. Based on information on the global tree cover in 2000, the yearly losses until 2009 and the gains until 2009 (assuming the growth between 2000 and 2012 that is given in the data set is linear), global tree cover in 2009 is estimated. We aggregated from the original 30 arcsec resolution to 0.5°. Regions with less than 5 % tree
cover north of 55° N are fixed as our study area (cf. Fig. 1). In Hansen et al.'s data, a tree is defined to have a minimum height of 5 m which is tall for circumpolar areas. The studied area will therefore include parts of the taiga-tundra transition zone, tundra as well as polar deserts.

The landscape in this study area exhibits complex microtopography caused by polygons and is characterized by abundant (thaw) lakes. Though vegetation often fully covers the ground (Stow et al., 2004), vascular plant cover is sparse and with one to
three months the growing season and carbon uptake period are short in high-latitude tundra. The RGB images from Sentinel-2 for selected spots in Fig. 1 give examples of what tundra landscapes and the tundra-taiga transition can look like. Further, for small areas on the Alaskan North Slope and the root of the Taimyr Peninsula (the corresponding places are indicated in Fig.1), Fig. 2 and Fig. 3 illustrate the mean annual cycles of environmental conditions together with Sentinel-2 images at given

points in time during the growing season. Temperatures rise above the zero degree Celsius line in late May and snow melt is often only completed in June. GLEAM soil moisture is usually highest at the time of the start of the growing season and also illumination is close to maximum. Temperatures keep increasing until July (ice layers on lakes can persist until July, cf. Fig. 1, example of tundra close to the Laptev Strait). GLEAM soil moisture is lowest at the time of highest temperatures, while
areas of open water are largest. Light conditions are already diminishing. Temperatures fall below the freezing point in late September or October. During this short period of favourable growing conditions vegetation phenology rapidly develops (cf. temporal sequences of RGBs in Fig. 2, 3 and Arneth et al., 2006).

For the evaluation of soil moisture we will focus the analysis on GLEAM data. GLEAM and AMSR-E/2 data on soil moisture partly show different qualitative behaviour (Fig. 2, 3, while they are more similar qualitatively in additional examples in Fig. A2
middle and bottom panel). Directions of possible explanations might be: 1) Both refer to slightly different quantities. AMSR-E/2 denotes the volumetric soil moisture in the uppermost 1 cm of land surface while GLEAM refers to surface soil moisture (denoted as 0-10 cm depth). Full comparability might thus not be given. 2) The applicability of GLEAM data for high latitudes has not been shown. 3) The fraction of open water from AMSR-E/2 shows surprising behaviour in that we expected the annual maximum to appear in early summer and not rather late in the summer. In the examples above of different behaviour between
GLEAM and AMSR-E/2 soil moisture, the latter shows similarly unexpected behaviour. This is indicative of the problems that still need to be faced for soil moisture retrievals in the high latitudes. 4) It is actually not clear what these quantities mean in tundra ecosystems. The microwave signal from the persistently wet peat base of the moss layer can penetrate the dry moss layers (which might reach a thickness of tens of centimetres), but the signal in GLEAM in similar cases is unclear.

## 3   Results

### 3.1   Timing of the annual peak in vegetation activity and greenness

The distribution of the timing of the annual maximum in polar treeless regions (Fig. 4, regionally and over years) shows a distinct order of the satellite vegetation proxies. All proxies indicate highest plant activity and biomass after the summer solstice. While APAR is highest around DOY191 (July, 10th), METGPP indicates maximum photosynthetic activity at a similar time, RSGPP four days later. It is the time when surface soil moisture is almost at minimum according to GLEAM (Fig. A3).
The SIF GFZ peaks only about one week later (four days later in case of SIF.daily.int GFZ) around DOY202 (July, 21th). The observations show that the SIF GFZ peak, potentially indicative of highest photosynthesis, is reached in close synchrony with the annual temperature peak. In fact, SIF GFZ is the only variable whose distribution of the peak timing is not significantly different from the one of temperature (paired two-sided t-test, significance level of 5%). SIF NASA on average peaks only on July, 28th (DOY 209), at a similar time when surface inundation by open water is highest (Fig. A3, although both the SIF
NASA and AMSR-E/2 data indicate a large range). Removing the effect of incoming radiation from the SIF measurement (by dividing by cos(sun zenith angle)) shifts the annual maximum for SIF.cosSZA GFZ compared to SIF GFZ by six days, and by eleven days for SIF.cosSZA NASA compared to SIF NASA and there is comparatively large scatter. The yearly maximum in vegetation indices occurs in late July/ early August, where EVI and NIRv peak around July, 31st (DOY212), one and a half

weeks after the temperature and SIF GFZ maxima. VOD peaks on average in close temporal agreement with EVI and NIRv. Finally, up to five days later the MODIS VIproduct as well the NDVI reach their maxima in the first week of August. Statistically, the NDVI peak timing is significantly different from the one of all other vegetation proxies (paired two-sided t-test, 5%). The scatter between years and regionally (standard deviation) is smallest for the model GPP data where interestingly the RS set-up shows a larger spread. APAR and the vegetation indices based on MCD43C4 have standard deviations of 12 to 14 days, slightly smaller than the ones of the MODIS VIproducts and of the SIF GFZ data sets (17 to 20 days). Largest uncertainties are shown by the SIF NASA data sets and VOD. Still, grouping indicators based on the similarity of their intrinsic properties (e.g. RSGPP and METGPP, SIF GFZ and SIF NASA, EVI and NIRv) shows such groups have a consistent behaviour and follow a certain pattern: APAR indices < model GPP < SIF < fAPAR (with EVI, NIRv,VOD < NDVI).

## 3.2 Spatial patterns of the annual maxima of the satellite vegetation proxies and of the lags between them

Overlaid on the general order of the different groups of proxies, there is considerable spatial variability in the timing of the maximum for each satellite indicator (Fig. 5). In areas close to the date line (i.e. easternmost Siberia and Alaska), most proxies peak slightly earlier than in northern Canada (mainland and islands) or the coasts of western and central Siberia (i.e. Taimyr Peninsula and regions of the Lena Delta and the Laptev Strait). In general, the spatial pattern of the timing of the annual maximum of the satellite indicators qualitatively closely corresponds to the dynamics seen in air temperature and partly in the surface soil moisture (GLEAM). Incoming light shows partly reversed patterns with earlier maximum irradiance in northern Canada and western-central Siberia.

We test whether the annual maximum is shifted systematically or whether there are spatial gradients in the peak lag between proxies, and plot maps of the average lag between the peaks of selected proxies and NDVI. We take the NDVI because it is the most widely used vegetation index for productivity studies, both from the satellite as well as in ground-based observations, particularly in polar tundra. Figure 6 confirms the general pattern of a shifted annual peak of NDVI as compared to NDVI.Rg, RSGPP, SIF GFZ, VOD and the similar timing like EVI all over the study area. NDVI.Rg, RSGPP, SIF GFZ, VOD and air temperature reach their annual maxima at significantly different times than the NDVI (Fig. A4). Figure 6 further shows that the lag is not homogeneous in space, but that it is largest in vast areas of polar desert in northern Canada, on Iceland and in the northern part of the Siberian Taimyr Peninsula. Interestingly, the tundra regions that exhibit the largest time difference of the annual maximum between the individual vegetation proxies and NDVI tend to correspond to those where the annual maximum is reached comparatively late (Fig. 5). In contrast to the denser vegetation cover on the Siberian coast, very sparse vegetation (i.e. devoid of shrubs/ woody vegetation) characterizes the northern Canadian regions, both the islands and the mainland, as well as the northern part of the Taimyr Peninsula (cf. Fig. A1, Walker et al. (2005,  their Fig. 1)). In addition, there is a comparatively high amount of lakes and high fractions of barren regions in the areas of large NDVI lags in the central Siberian coastal areas (close to the Laptev Strait), coastal Alaskan North Slope and in mainland Canada northwest of Hudson Bay. Similar results are obtained for EVI.Rg, METGPP, SIF.daily.int GFZ (Fig. A5). Next to these general observations, VOD indicates a much later peak in smaller, but contiguous areas in northwestern Canada as well as in lake-rich regions on the Siberian coast (again

the coast close to Laptev Strait). NDVI lags to METGPP are generally slightly larger than to RSGPP. No outstanding region emerges for the lags compared to SIF NASA. The illumination correction of SIF (SIF.cosSZA) reduces the time difference to the NDVI compared to the instantaneous SIF as expected.

### 3.3  Spatial patterns of peak timing and of peak lags to the NDVI in relation to environmental variables

Putting the annual maximum of the NDVI into relation with the one of the environmental variables, partly similar spatial patterns emerge like for the vegetation proxies (Fig. 6). Precisely, air temperature and soil moisture (GLEAM minimum) peak earlier everywhere and have the largest time difference to the NDVI in the sparsely vegetated areas on Iceland, northern Canada and parts of the Taimyr. Conversely for the microwave retrievals of the amount of open water on the surface, where mixed temporal relationships with the NDVI peak are observed. Most water as mapped from microwave remote sensing is present after the time of NDVI peak in large parts of northern Canada, Greenland and in land masses close to the date line, while NDVI is at maximum after the fraction of open water in all other regions. Summer precipitation might increase surface wetness and influence the microwave fraction of open water as typically highest surface inundation is expected to happen immediately after snow melt.

It is further interesting to test whether a certain temporal relationship between the maximum in photosynthesis or greenness and the dynamics of the environmental conditions holds across years. In Figure 7, the environmental variable with the highest absolute value of the rank correlation with the timing of the maximum of the given satellite proxy across years and in a spatial moving window is displayed. The important role of energy-related variables, mostly temperature, for vegetation activity and growth is highlighted by widespread highest correlations with temperature and radiation for RSGPP, SIF GFZ, EVI and NDVI, while moisture-related variables are most important only in about one third of the pixels. VOD and NDVI.Rg do show a strong relationship with the annual temperature maximum only in 25 to 30% of the pixels and about half of the pixels show a higher importance of soil moisture or open water on the surface. Worth to note is that the contiguous regions with higher relationships with moisture related variables for RSGPP, EVI and NDVI are situated in northwestern Canada and parts of the Taimyr (Fig. A6).

Interestingly, apart of the strong correlative relationship of photosynthesis with temperature, the lags in annual peak timing between temperature and SIF GFZ (Fig. A7) are barely statistically significant. Peak lags between RSGPP and temperature are only significant in northern Canada and easternmost Siberia. Conversely, EVI, NDVI, VOD and in the largest parts of the study area also NDVI.Rg peak at significantly different times in a year than air temperature.

### 3.4  Consistency of the annual peak lags between different land covers and across years

The fact that there is spatial variability in the shift between the annual peaks of the satellite proxies relative to the NDVI suggests that the proxies differ in how strongly they indicate the spatial gradients in the peak DOY. We test to what extent the shift of the annual maximum holds across different years and whether there is a dependency on the land cover. Figure 8 (and Fig. A8 but here based on the actual time series and not the bootstrapped values) shows the peak lags as a function of land

cover based on ESA CCI and for all years in the study period separately. Peak lags to the NDVI per proxy are generally similar between land covers, although there is a tendency in several proxies (excluding the VIproducts of MxD13C1, METGPP, SIF NASA) for larger lags in regions classified as moss. According to Fig. A1, this largely corresponds to the sparsely vegetated areas in northern Canada with also high cover fractions of water and barren. The smallest lags of RSGPP and SIF GFZ are shown for mixed shrub-tree land covers. There is also some variability between years which is largest for the timing of the moisture related variables of maximum extent of open water and GLEAM soil moisture minimum. Conversely, variability of the peak lag per land cover is smaller for the vegetation proxies between years. VOD is maximized five to seven days earlier than NDVI in tundra-like vegetation types, but has no consistent relationship with NDVI in forests and water. Interesting patterns are the large NDVI-t2m lags in 2015 in grass-herbs and 2012 in moss as well as the particularly small lag in 2013 in mixed shrub-tree areas. This behaviour between years is qualitatively similar between RSGPP-NDVI and SIF GFZ-NDVI and rather caused by variability in the peak timing of temperature than of the NDVI (not shown).

## 4   Discussion

Despite the considerable challenges for remote sensing applications in high latitudes and inherent comparatively large variability in the timing of the annual maximum, the differences in the peak timing of families or groups of key satellite indicators of plant productivity appear to be ordered in polar tundra. Absorbed energy (APAR, both EVI.Rg and NDVI.Rg) is maximized roughly one month after peak irradiance in early July. Regarding model GPP and SIF as indicators of photosynthetic activity, there is a time lag between them of four days to two and a half weeks, depending on the combination of data sets. Model GPP peaks at a similar time like APAR. SIF GFZ reaches maximum one to one and a half weeks after (July, 21st) at a very similar time like air temperature, but SIF NASA only in the end of July (DOY 209, July, 28th). Greenness (EVI and NIRv) culminates three weeks after APAR and one and a half weeks after SIF GFZ. NDVI maximum is delayed on average three more days. Peak vegetation water content is indicated by VOD at a similar time like EVI and NIRv. The spread in space and across years is however large.

Vegetation activity is highly (though not exclusively) temperature-driven in tundra  (e.g. Jia et al. in  Stow et al., 2004; May et al., 2017; Chapin, 1987). In the beginning of the growing season, light is abundant and plants rely on rhizome nutrient and carbohydrate reserves to rapidly increase photosynthetic activity (Arneth et al., 2006) and growth (Chapin, 1987) by exposed mosses, lichens and evergreens after snow melt and rapid leaf out of deciduous plants. The fact that the photosynthesis seasonal maximum is reached in close and significant (SIF GFZ and RSGPP) temporal agreement with air temperature adds plausibility to the observed patterns in model GPP and SIF. As the time of favourable environmental conditions for growth is short, several plant types strongly invest into their photosynthetic capacity until late in the growing season to make use of the available light and temperature (Rogers et al., 2017). At the time when greenness is at maximum, photosynthetic rates are decreasing as PAR is already strongly reduced and also the temperature peak has passed. The peak timing of SIF before greenness might hence indicate that although photosynthetic potential (fAPAR) is not yet fully developed, plants profit from the still higher amounts of

light and maximal temperature in the year to reach peak photosynthetic rates in the second half of July. Prolonged investment of photosynthates into plant tissue results in a delayed maximum of green biomass. The widespread agreement between green-ness proxies and photosynthesis proxies in high correlations with the temperature maximum across years and space (Fig. 7) are also indicative of the coordinated dynamics of annual maximum photosynthetic activity and the resulting peak photosynthetic

potential (fAPAR) with temperature.

The results of Chadburn et al. (2017) support this interpretation of the patterns in the satellite observations: In their site-level evaluation of carbon fluxes in the high latitudes in Earth system models, GPP always depends on temperature, but is limited rather by LAI in the first part of the growing season until the end of July. After that, GPP is more driven by light, which qualitatively agrees with the earlier photosynthesis peak seen in our results. Besides, at the site-level, gas flux measurements

find a similar timing of maximum GPP in the first half/mid-July (Emmerton et al., 2016) and at the time of the annual temper-ature peak (Kross et al., 2014; Welker et al., 2004). Similarly for Lafleur and Humphreys (2008) who report on largest annual site-level NEE after summer solstice near the annual temperature maximum between DOYs190–210 and a dominant role of GEP in driving these dynamics.

An interesting aspect of the general time lags between proxies is the ten to twelve days time difference between peak APAR and peak SIF GFZ. According to the Monteith model for SIF, the observation of a time difference of peak APAR and peak SIF suggests that SIF might contain information on temporal dynamics of actual photosynthetic light-use-efficiency of tundra vegetation. Circumarctic vegetation is adapted to low light intensities to allow photosynthesis also at low irradiance (Chapin, 1987; Rogers et al., 2017, and references therein). Consequently, photosynthesis will rapidly become light-saturated, a situa-

tion that calls for high levels of non-photochemical quenching in order to avoid photodamage and inhibition by excess energy. Under these conditions, the efficiencies of carbon fixation and fluorescence emission are positively correlated (Porcar-Castell et al., 2014). Consequently, our results indicate a potential benefit of using also SIF in modelling photosynthetic carbon uptake in circumpolar tundra for its apparent sensitivity to both APAR and photosynthetic light-use-efficiency. Although they did not report on results on GPP, Luus et al. (2017) found higher agreement between modelled NEE and eddy-covariance derived NEE

when phenology is prescribed by SIF instead of EVI in tundra in Alaska.

Although model GPP, SIF GFZ and SIF NASA are indicators of photosynthetic activity and they peak closer to the annual temperature maximum than both APAR and greenness indices, they do differ in peak timing as well. In the following we test a suite of possible explanations from physiological decoupling between SIF and GPP to artefacts originating from data process-

ing and data characteristics.

1) The time difference of the yearly maximum of model GPP and SIF GFZ cannot be fully explained. Since the SIF maximum is reached in close and statistically significant temporal agreement with air temperature, it might indicate that SIF shows higher sensitivity of photosynthetic rates to temperature. The earlier peak of model GPP might be explained by a possible higher sen-sitivity to radiation as it is challenging to model effects of water table depth or temperature acclimation. This is especially true

for the METGPP that culminates slightly earlier than RSGPP and that is driven by a mean seasonal cycle and not temporally

resolved greenness. Generally, model performance of model GPP is reduced in extreme climates (Tramontana et al., 2016). Furthermore, FLUXCOM GPP might not accurately represent GPP in tundra due to the small size of training data. FLUXCOM GPP is trained at FLUXNET sites and according to Tramontana et al. (2016) there are eleven sites north of 55° that are not classified as forest or temperate and serve the modelling of GPP in our study area. Five of them are located north of 65° and the three training sites classified as Arctic are all located in Alaska.

We attempted to get an indication of whether - compared to *in-situ* eddy-covariance-derived GPP and site-level temperature - there is a systematic shift of either model GPP (for the 0.5° grid box whose centre coordinates are closest to the flux tower) or SIF GFZ (16-day averages of all GOME-2 footprints with centre coordinates within a radius of 30 km from the flux tower). Such comparisons of satellite observations with site-level measurements can be very helpful for the interpretation of the patterns seen from space, but here they remain inconclusive. There is agreement between site-level and regional analysis with respect to the order of model GPP before temperature, but reversed order regarding the time of peak SIF GFZ and temperature (SIF GFZ earlier at site-level, but after temperature at regional scale, Fig. A9, A10). The comparison is hampered by a number of issues: i) mismatch of scale and spatial representativeness, ii) limited temporal overlap of site-data with satellite records, iii) data quality of NEE and temperature at site-level (e.g. Fig. A11, A14), iv) quality of partitioning at site-level (partly strongly different behaviour between partitioning methods, Fig. A12, A14, though they mostly converge versus the peak growing season, see also Runkle et al., 2013; Parazoo et al., 2018), v) temporal scale of daily integrated EC measurements and instantaneous SIF observations in the morning (Parazoo et al., 2018, find similar effects in studying the start of the growing season). These issues clearly suggest that EC cannot always be used as the 'truth' in evaluating satellite observations. At the same time these results strongly underline the observational problems in tundra explained in the introduction and call for more *in-situ* measurements that are well characterized and understood in order to interpret the signals seen from the satellite.

As we do not find a physiological explanation, artefacts in SIF might originate from seasonal cloud cover as well as in satellite overpass time and explain the time difference in annual peak between model GPP and SIF GFZ. Clouds affect SIF values both physiologically at the leaf level and on its way from the canopy to the satellite. A possible bias of rather clear-sky instantaneous observations of SIF in the morning hours compared to model GPP as an all-day measure might occur (Parazoo et al., 2018). Although there have been no tests in tundra, in both model simulations (Frankenberg et al., 2012) and empirical analyses (Köhler et al., 2015) choosing different thresholds of cloud cover does not strongly affect temporal patterns of SIF as a large fraction can still be detected at moderately thick clouds. In our tests (Fig. A15), stricter cloud filters tend to slightly enlarge the lag between model GPP and SIF GFZ, while SIF NASA shows no change. Conversely, SIF GFZ peak time shows no sensitivity to overpass time, while in SIF NASA noon-overpasses indicate a slightly earlier peak than using measurements between 8 and 14 A.M.. Although data availability becomes problematic in the case of OCO-2, resulting in discontinuous climatologies , there is largely agreement with OCO-2 SIF when averaged of larger regions (Fig. A17). This suggests that the peak timing obtained from GOME-2 SIF GFZ observations is reliable.

2) There is also a relatively large inconsistency between SIF GFZ and SIF NASA. We conducted tests in that we strictly filtered for the same cloud fractions (0.3) and overpass times (8-14 LST) in both data sets. We did not attempt a one to one

allocation of individual soundings between data sets and it needs to be noted that the cloud fraction is defined in a slightly different way between SIF GFZ and SIF NASA (see methods). Indeed, a stricter cloud filter of 0.3 instead of 0.5 delays the SIF GFZ peak slightly, bringing it into closer but not full temporal agreement with the SIF NASA peak (Fig. A15). Strict filtering for midday over pass times advances (as expected, Parazoo et al., 2018) the annual peak, but for other combinations of overpass filters no systematics appear.

We argue that the NASA data set is more prone to noise (for example for retrievals over bright surfaces when there is partial snow cover) due to the generally lower absolute values that result from a narrower retrieval window and that this severely affects the identification of the annual peak. This is indicated by the large spread in Fig. 4, by the less pronounced spatial patterns in Fig. 6, and by the time series examples in Fig. A16. Further, the illumination correction amplifies noise in the time series. This is thus an example for the degradation of the signal by the division by cos(SZA) and calls for caution in applying it. We conclude that the different cloud fractions can explain a part of the difference between the SIF GFZ and SIF NASA and argue that the remaining difference might be ascribed to the different noise levels in the two data sets that make a reliable peak identification in SIF NASA more difficult than in SIF GFZ and also lead to a larger spread.

In our results, NDVI and the MODIS VIproducts are the latest greenness proxies and peak around DOY216 (August, 4th). The NDVI is a widely used indicator of productivity and comparing to ground-based measurements as well as satellite observations with the AVHRR instrument shows mostly support for NDVI peak in very late July or the beginning of August. Ground NDVI along a transect in Alaska by Huemmrich et al. in Stow et al. (2004) agree with the MODIS NDVI in that the seasonal maximum is observed at DOY218 in the beginning of August. In a second year there is even a second peak at DOY230 in ground-based NDVI. Huemmrich et al. (2010a) show time series of ground-based NDVI in Alaska that reaches the peak about two weeks earlier (at DOY203) than the average MODIS NDVI in our results but remains high until the end of August. However, May et al. (2017) report on peak dates of in-situ measured NDVI in Alaska roughly one week to two weeks earlier (DOY 199-207) and tell about the beginning of senescence after the first sunset in late July or the beginning of August. Finally, satellite-based bi-weekly NDVI from the AVHRR instrument is shown to peak between July, 22nd and August, 4th (Jia et al. in Stow et al., 2004; Zhou et al., 2001). Still, the onset and the peak timing of MODIS-based NDVI has also been found to not be consistent with ground based observations of NDVI (Gamon et al., 2013) which might suggest partly questionable reliability of satellite NDVI.

While the reported ground observations were all conducted in Alaska, in our circumpolar results the NDVI largely agrees with the other vegetation indices EVI and NIRv and only peaks later in the northeasternmost parts of Canada (Fig. 6 and A5). In addition to the Taimyr and coastal North Slope Alaska, these are the same regions where also the NDVI lags to all other proxies are largest. According to the ESA CCI land cover, those regions are characterized by moss (Fig. 1). Moss often has no clear seasonal cycle in greenness making a peak identification difficult. Moreover, vegetation is particularly sparse in the form of prostrate dwarf shrubs and there are extensive barren areas with rich lake cover in those northern Canadian areas (Fig. A1, Walker et al., 2005, their Fig. 1, 2e and 3). This renders the reflectance based observation particularly sensitive to background

conditions, especially without a clear seasonality in greenness (Walker et al., 2005, their Fig. 2f). Confirmation for this hypothesis of strong contamination of the NDVI signal is given by the sharp transition from the very large lags in northeastern mainland Canada (eastern Barren Grounds) to lower albeit still negative lags to the northwestern part of mainland Canada (Fig. 6 and A5, corresponding to the land cover transition between bare-sparse in the western parts of the Barren Grounds to moss in the more easterly regions of the Barren Grounds in Fig. 1). Similar like the northern Canadian islands and northeastern mainland, northwestern Canada is characterized by many lakes (Walker et al., 2005) and ESA CCI land cover reports on sparse vegetation with much moss and open water as well (Fig. A1). However, in these more western areas, vegetation changes to rather erect dwarf shrubs and graminoids (Walker et al., 2005, their Fig. 3) which exhibit a clearer seasonality (Fig. A18, panel of northwestern Canada) than the very sparse vegetation in the eastern parts with prostrate shrubs (Fig. A18, Canadian Archipelago (northern mainland and islands), northeastern Canada and Iceland). Those are also less affected by increasing values at the beginning and at the end of the growing season that are partly even higher than the summer maximum and severely affect the identification of the annual peak. We speculate that possible explanations for this might be an increasing effect of low SZA late in the growing season (Kobayashi et al., 2016) affecting low NDVI in particularly sparse vegetation heavily. NDVI might also be strongly decreased by standing surface water (Gamon et al., 2013) from snow melt or intermittent precipitation that has not yet drained or evaporated until later in the growing season. Only upon drying, will the NDVI increase due to the missing water absorption of the NIR, and this might affect the trajectory of NDVI strongest in the sparsely vegetated regions with the largest peak lags.

The highest vegetation water content is indicated by the VOD at a very similar time like the peak values of the vegetation indices - which might corroborate the usefulness of VOD to indicate vegetation biomass also in tundra ecosystems. Both vegetation indices and VOD are sensitive to vegetation structure and density, VOD in addition to water content (Liu et al., 2011). Especially in low biomass regions - as applied for tundra - a linear relationship between VOD and vegetation water content has been found (Teubner et al., 2018). The similarity of VOD as an indicator of total aboveground biomass to the vegetation indices might also support our interpretation of delayed peak greenness compared to photosynthesis due to longer lasting build-up of plant material and pigments and indicate that the proposed relationship by Teubner et al. (2018) of VOD-vegetation water content-LAI to GPP does not fully hold in tundra. Conversely, when looking at examples of the mean seasonality of VOD in comparison to the other vegetation proxies for selected regions (Fig. A19) the VOD annual cycle appears broader and the peak less well defined than the one of the vegetation indices. This could indicate that the vegetation water content changes only slightly during the growing season while possibly the chlorophyll concentrations independently exhibit more pronounced dynamics and affect the vegetation indices more strongly. Another possible factor contributing to the broader annual cycle is VOD being related to the water content of the *total* aboveground biomass (as opposed to green or photosynthetic biomass), including moss and woody components and litter. If persistently wet, moss might drive the VOD signal but less strongly the greenness signal. There is a high sensitivity of moss to air humidity as a consequence of the absence of roots. Despite a wet peat layer there might be several centimetres of dry moss material. In grass lands, consistently high emissions and consequently a lower VOD seasonality were ascribed to the contributions of litter and wet vegetation components (Grant et al., 2016). Similar

mechanisms might also hold for the VOD observations in tundra, especially considering that carbon turnover is slow in the high latitudes. It is also not clear what the VOD signal over water saturated soil might be. Moreover, the retrieval of VOD is strongly dependent on the representation of open water and soil moisture. Considering that the retrieval algorithms have not been calibrated for tundra-like conditions and that with the high heterogeneity regarding plant types and landscape components it might be difficult to accurately separate the contributions of vegetation from soil and water, it is not clear what VOD in tundra means. In future studies it would be useful to analyse a suite of different VOD products from different sensors and wavelengths together with in-situ observations in order to understand whether greenness and vegetation water content are strongly coupled in tundra or not and to what extent different retrievals affect the result.

Overall it needs to be stated that gaps in the data and the short growing season with often small seasonality and high noise levels challenge the reliable identification of phenological dates in all data sets.

## 5   Conclusions

We analysed and compared satellite-based indicators of plant productivity with respect to the timing of their maximum in Arctic treeless regions. Over the whole study area, peak productivity is generally reached in July with an order of APAR culminating in the first half of July together with model GPP followed by SIF GFZ one week later in synchrony with highest annual temperatures. SIF NASA is delayed by one week. EVI and NIRv indicate maximum greenness in the end of July, together with VOD as a proxy for vegetation water content. NDVI and MODIS VIproducts peak only in the first week of August. We interpret this sequence as an investment into growth of leaf tissue and pigments also after optimal conditions for assimilation regarding light and temperature have passed. Peak photosynthesis occurs earlier at a time when full photosynthetic potential has not yet developed but when light is still abundant and temperature favourable. Largest lags between NDVI and photosynthesis indicators are found in regions with particularly sparse vegetation without a clear seasonality in spectral reflectance that can heavily be confounded by low sun angles and the high abundance of lakes.

To our knowledge, only few studies of tundra vegetation have been based on other observables than spectral reflectance (Luus et al., 2017; Parazoo et al., 2018). A-priori it was questionable whether current satellite-based SIF data sets are useful for tundra vegetation considering the very large footprints, high susceptibility to noise and very small signals from the sparse vegetation. However, the spatial patterns of peak productivity of SIF are qualitatively similar to the ones seen in model GPP and reflectance-based observations. Furthermore, the fact that the SIF maximum is reached in close temporal agreement with air temperature might indicate a benefit for photosynthesis from highest temperatures. The general time difference between proxies of APAR and SIF suggest that there is information on light-use-efficiency contained in the SIF observations. Still, further studies are needed to verify this. The results of our study confirm the important separation between indicators of greenness and photosynthesis and non-negligible differences between data sets of the same indicators. Upon data availability in the future, similar cross-comparisons to the chlorophyll-carotenoid index (Gamon et al., 2016) and the photochemical reflectance

index (Gamon et al., 1992) in tundra might add yet additional complementary information on circumpolar vegetation dynamics.

*Code and data availability.* Code and data available upon request.

*Competing interests.* The authors declare no competing interests.

5 *Acknowledgements.* We thank Guido Ceccherini e Fabio Cresto-Aleina for help on polarstereographic plotting as well as Alessandro Cescatti and Mirco Migliavacca for discussion; Ulrich Weber for processing of GlobeLand30 data and FLUXCOM data; and Ramdane Alkama for processing Hansen forest cover data.

This work used eddy covariance data acquired and shared by the FLUXNET community, including these networks: AmeriFlux, AfriFlux, AsiaFlux, CarboAfrica, CarboEuropeIP, CarboItaly, CarboMont, ChinaFlux, Fluxnet-Canada, GreenGrass, ICOS, KoFlux, LBA, NECC,
10 OzFlux-TERN, TCOS-Siberia, and USCCC. The ERA-Interim reanalysis data are provided by ECMWF and processed by LSCE. The FLUXNET eddy covariance data processing and harmonization was carried out by the European Fluxes Database Cluster, AmeriFlux Management Project, and Fluxdata project of FLUXNET, with the support of CDIAC and ICOS Ecosystem Thematic Center, and the OzFlux, ChinaFlux and AsiaFlux offices.

We appreciate the permission for usage of eddy covariance-derived GPP and *in-situ* temperature measurements for the site RU-Ch2 which
15 have been provided by Mathias Göckede, Max-Planck-Institute for Biogeochemistry, Jena, Germany.

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

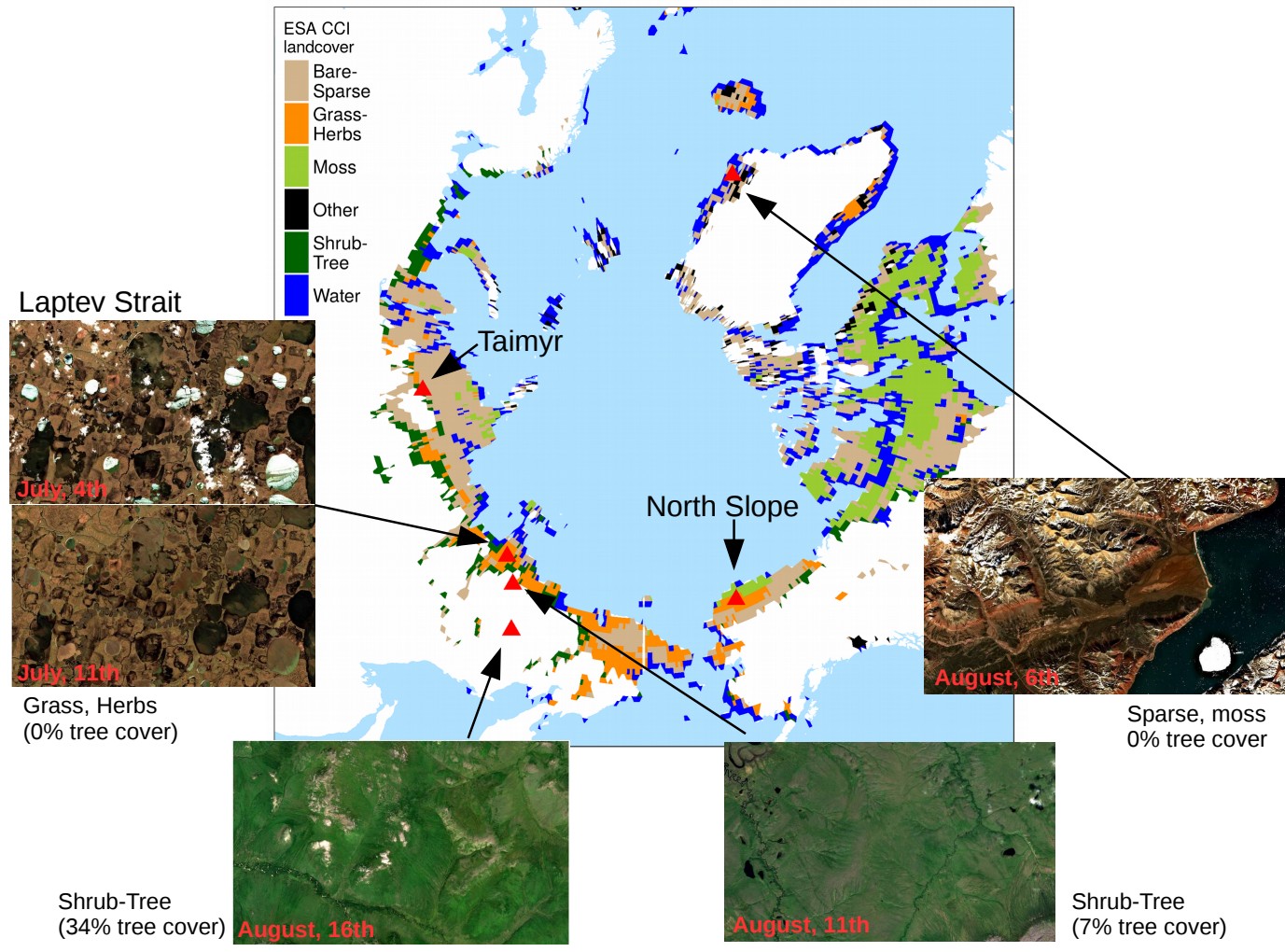

**Figure 1.** ESA CCI land cover in regions with less than 5% tree cover according to Hansen et al. (2013). Atmospherically corrected true colour images are from Sentinel-2 taken at different dates in 2017. For the region shown in each image the majority land cover is given and the tree cover percentage according to Hansen et al. (2013).

# Taimyr Peninsula

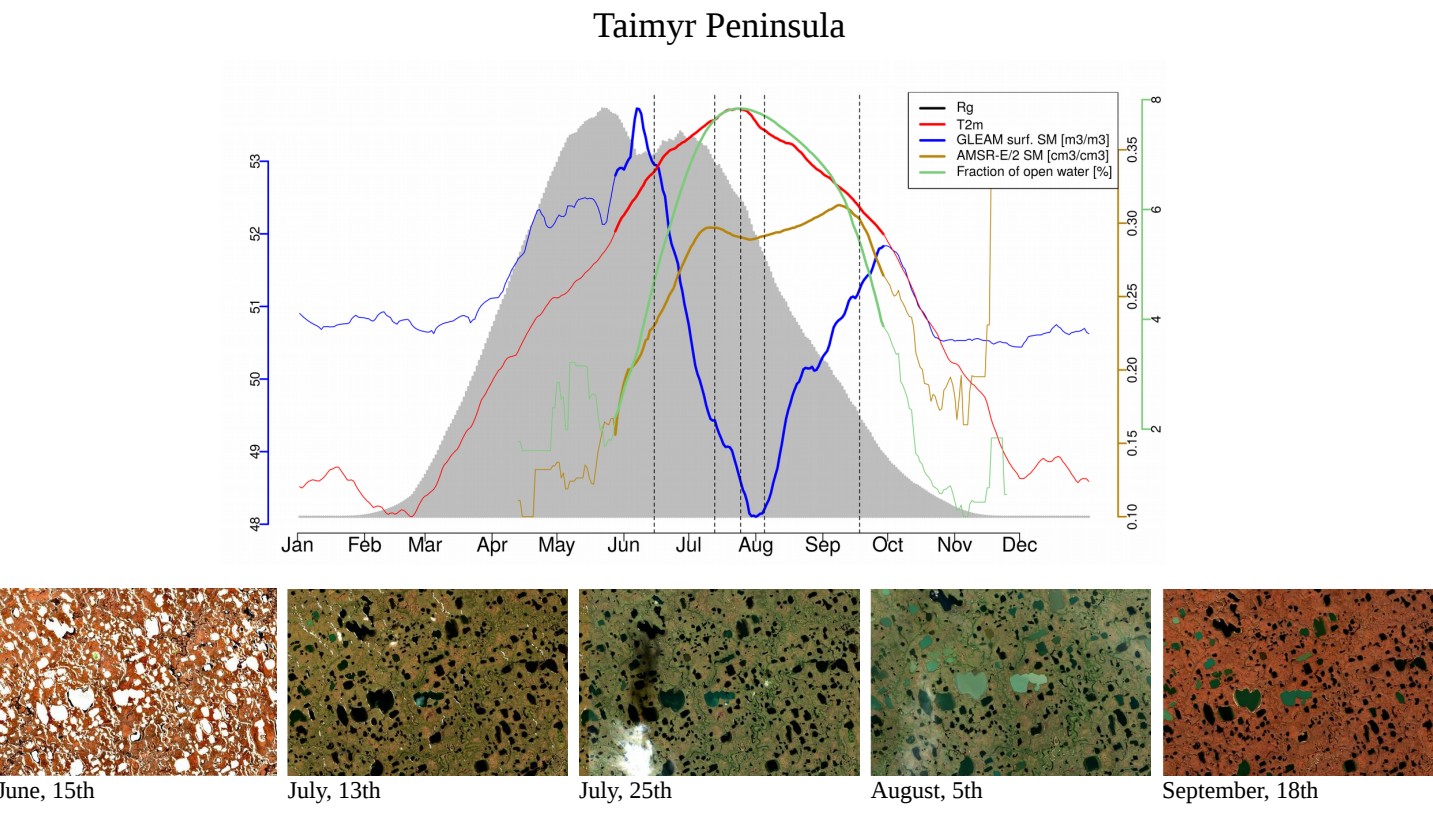

| June, 15th | July, 13th | July, 25th | August, 5th | September, 18th |

**Figure 2.** Climatologies of different atmospheric and land surface variables for a small area on the Taimyr Peninsula/ Russia indicated in Fig. 1. Bold lines indicate the time period when air temperatures are above the freezing point. Vertical dashed lines indicate the time of the year when the Sentinel-2 images shown in the second panel were taken. Sentinel-2 images are atmospherically corrected and taken in 2017.

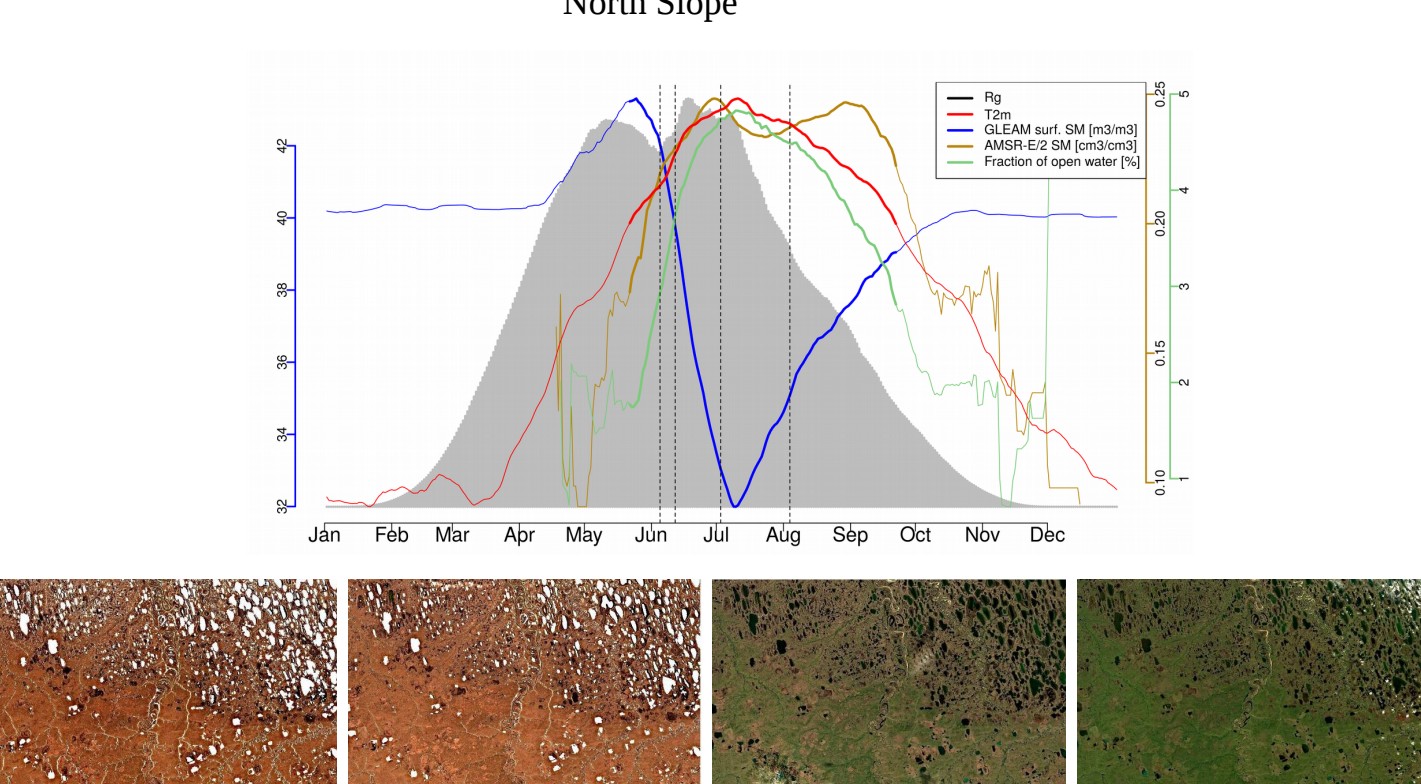

**North Slope**

June, 5th        June, 12th        July, 3rd        August, 4th

**Figure 3.** Climatologies of different atmospheric and land surface variables for a small area in the North Slope/ Alaska indicated in Fig. 1. Bold lines indicate the time period when air temperatures are above the freezing point. Vertical dashed lines indicate the time of the year when the Sentinel-2 images shown in the second panel were taken. Sentinel-2 images are atmospherically corrected and taken in 2017.

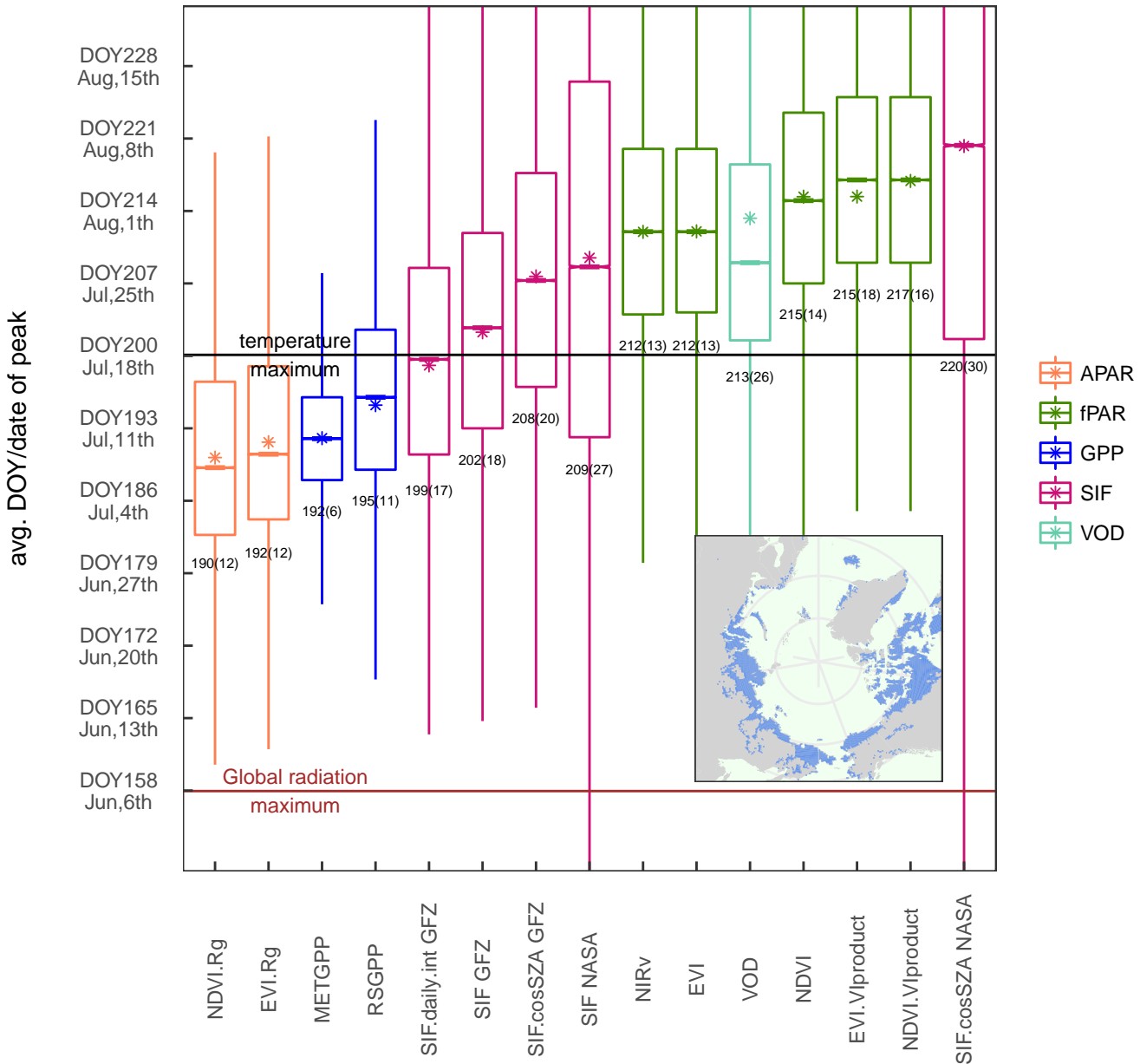

**Figure 4.** Distribution of the DOY of the peak of the different vegetation proxies over the study region and between years (spatial sampling matched between data sets for each year). Bars in the boxes indicate the median, stars the mean, the numbers below the bars denote the spatio-temporal mean (standard deviation). Colours of the bars denote grouping of the different variables according to the families of fAPAR, APAR, model GPP, SIF, VOD.

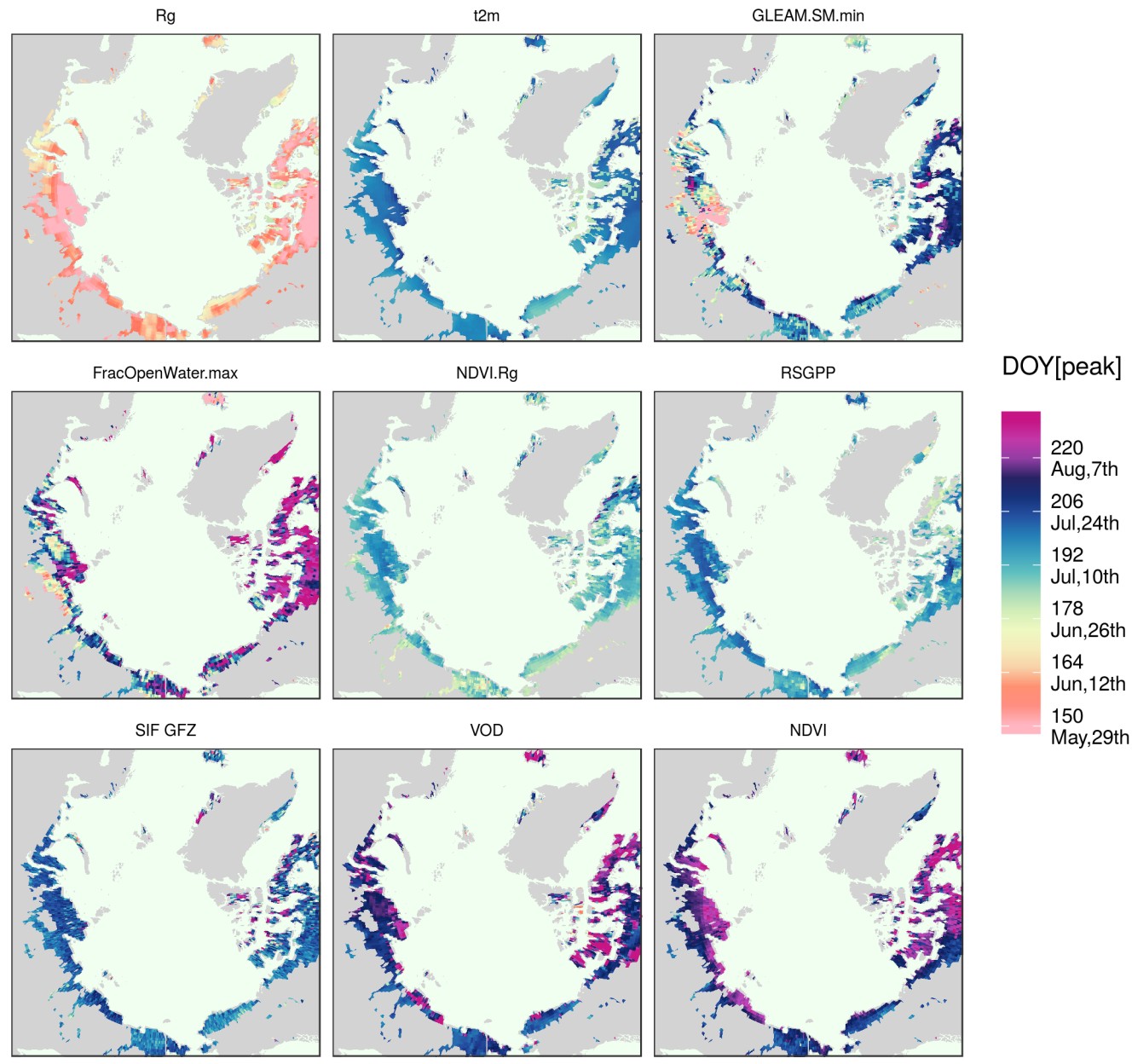

**Figure 5.** DOY of the annual maximum averaged over all years as indicated by selected vegetation proxies.

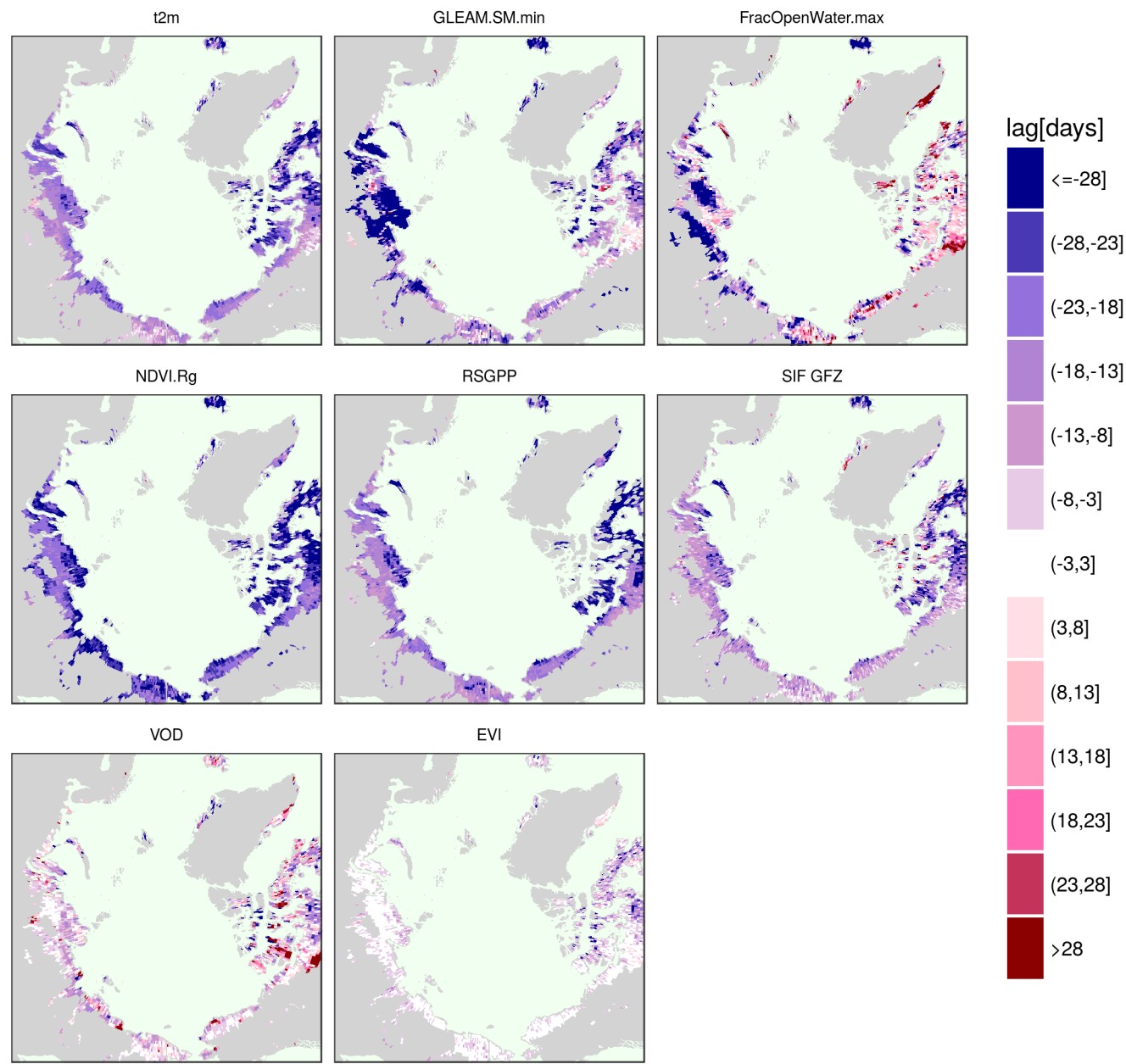

**Figure 6.** Average time difference of the maximum across years of selected vegetation proxies and the NDVI.

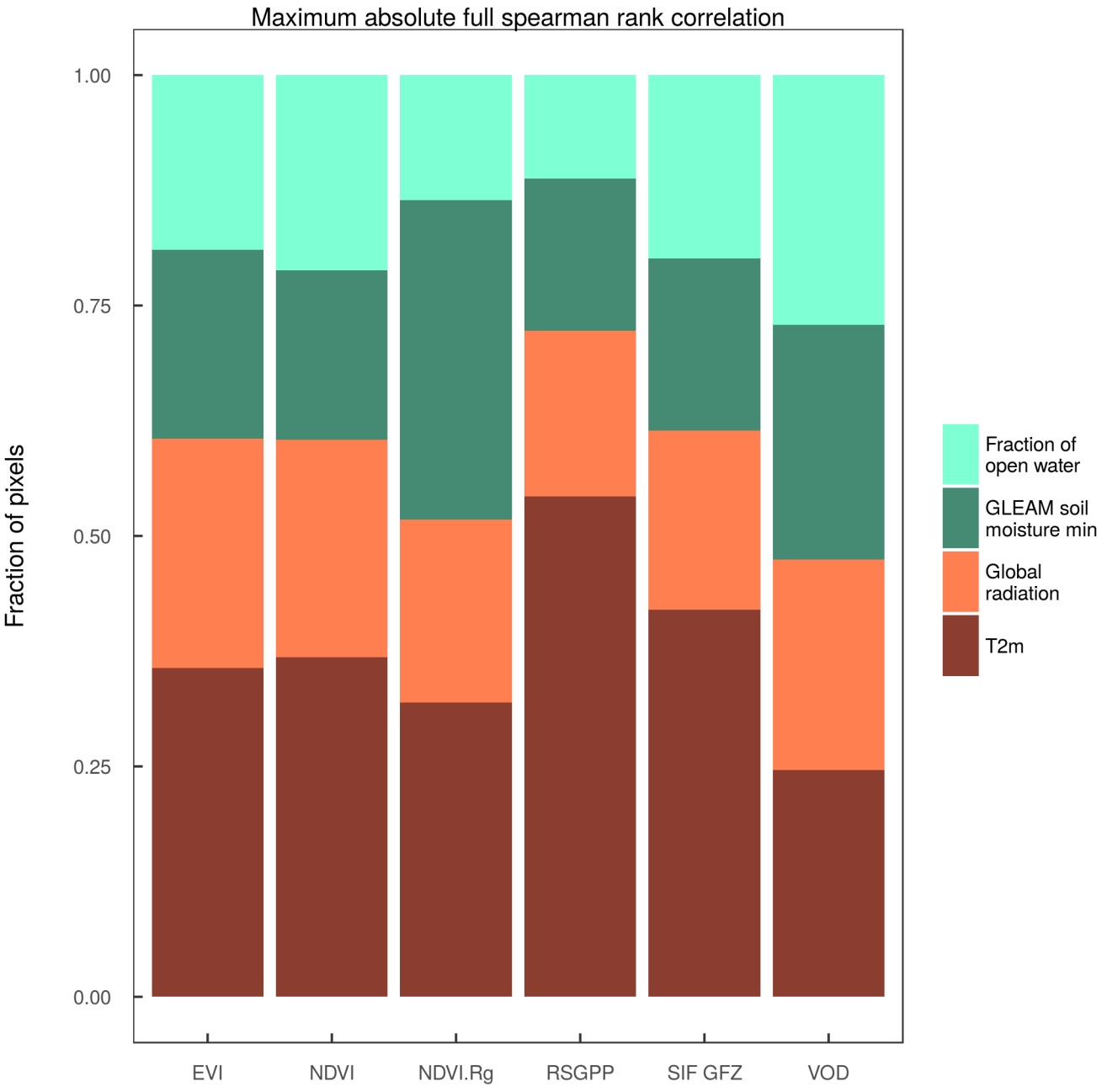

**Figure 7.** Frequency distribution of pixelwise spearman rank correlation between the peak DOY of the vegetation proxies across years and the peak DOY of environmental variables in spatial moving windows of 1.5° (so 9 spatial pixels times 10 years at most, correlated only if more than 20 data points available). Plotted here is the variable with the highest absolute correlation without filter for statistical significance. Full correlations have been calculated (no partial correlations). The corresponding spatial distribution of the maximum correlations is given in Fig. A6.

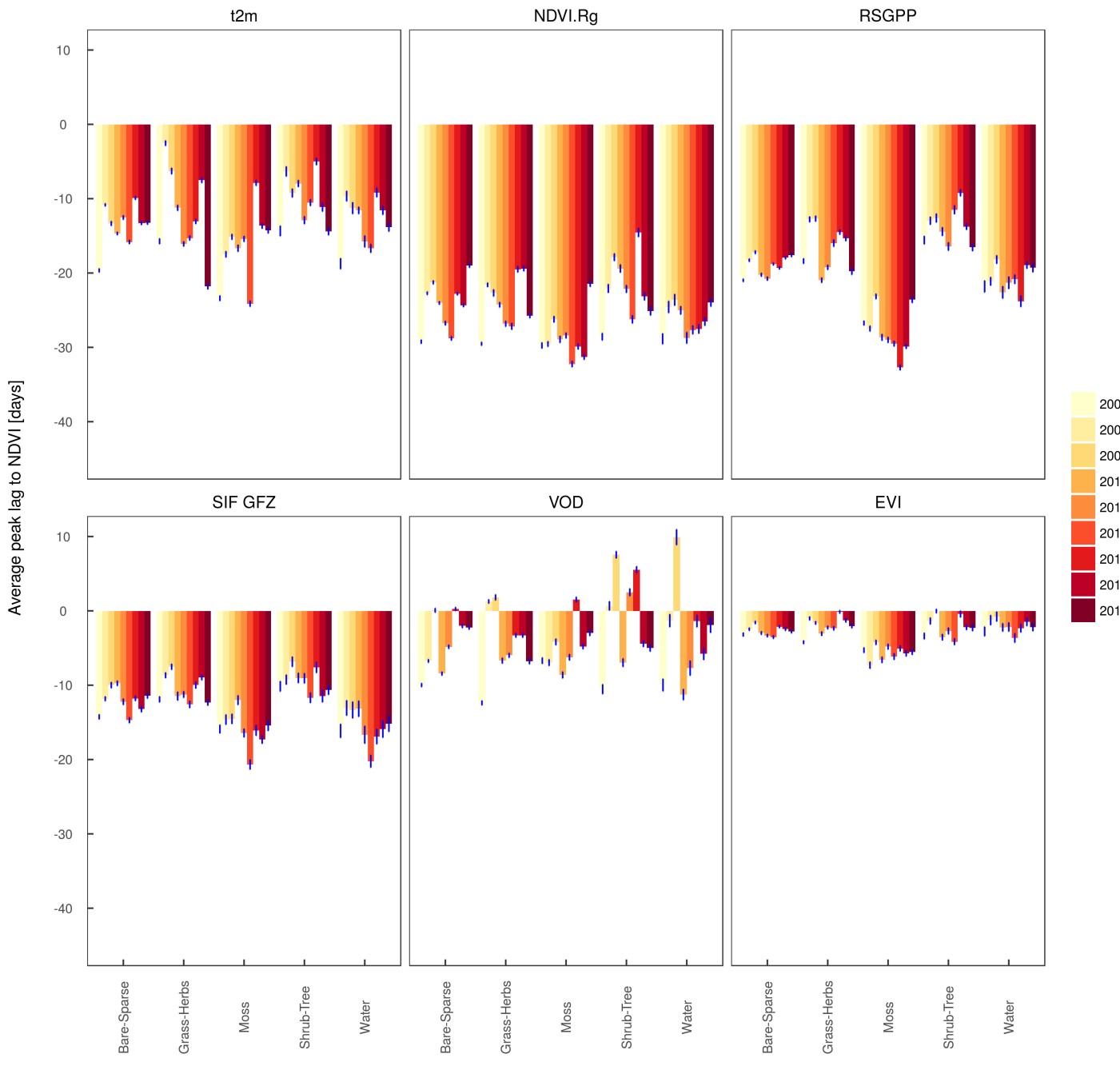

**Figure 8.** Average of the time difference between the peaks of one selected variable per family (APAR, fAPAR, SIF, GPP, VOD) and the NDVI as a reference, weighted with fractional cover per vegetation type (based on ESA CCI) and per year. Each bar represents the average over 50 bootstrap samples of the time series per year, pixel and vegetation proxy, error bars indicate 1.96 multiple of the standard deviation across the 50 samples.

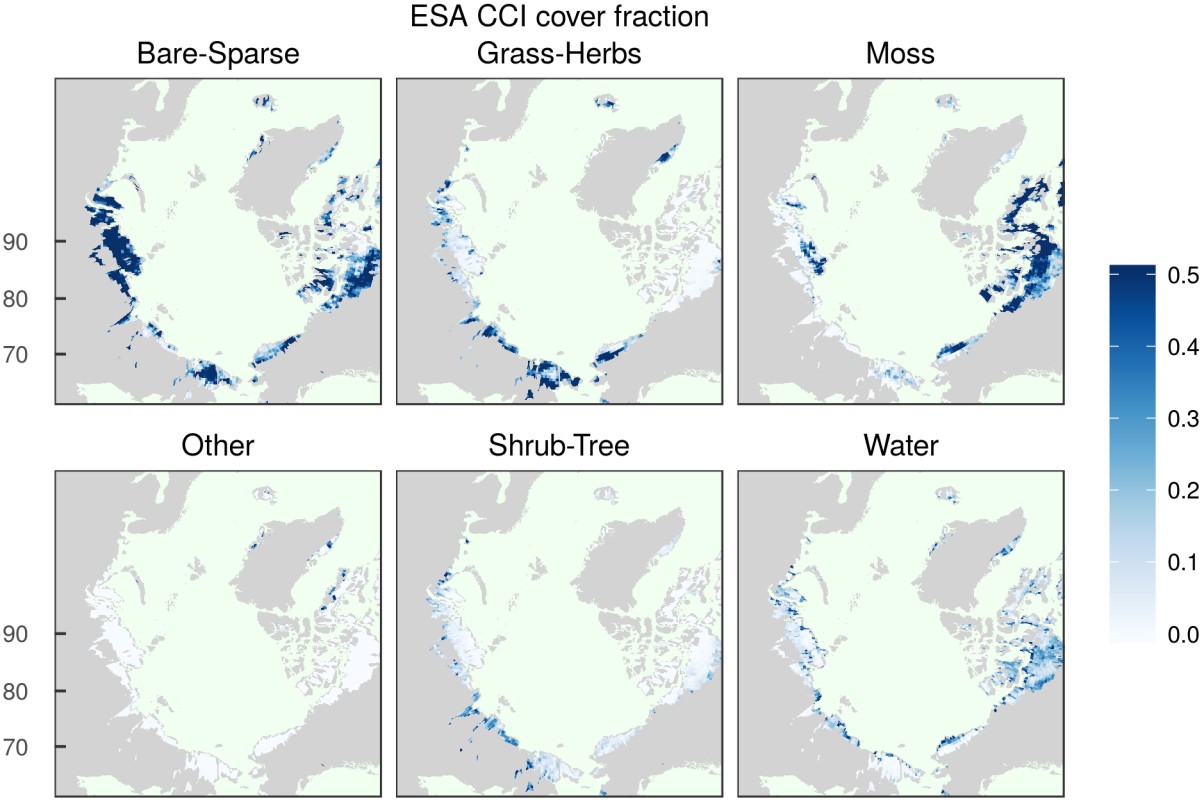

**Figure A1.** Fractions of the aggregated land cover classes for the ESA CCI land cover data set. The aggregated classes comprise 'moss' (class 100 in the ESA CCI classification), 'bare/ sparse' (classes 28-30,35-37, fractions of 16 and 19), 'grass/ herbaceous' (classes 26, fractions of 13,16,19,21–25, 33), 'woody' (shrubs and trees, classes 10–12,14,15,17,18,20, fractions of 13,16,19,21–25,31–33), 'water' (class 38, fractions of 31–33) and 'other' (remaining classes and fractions).

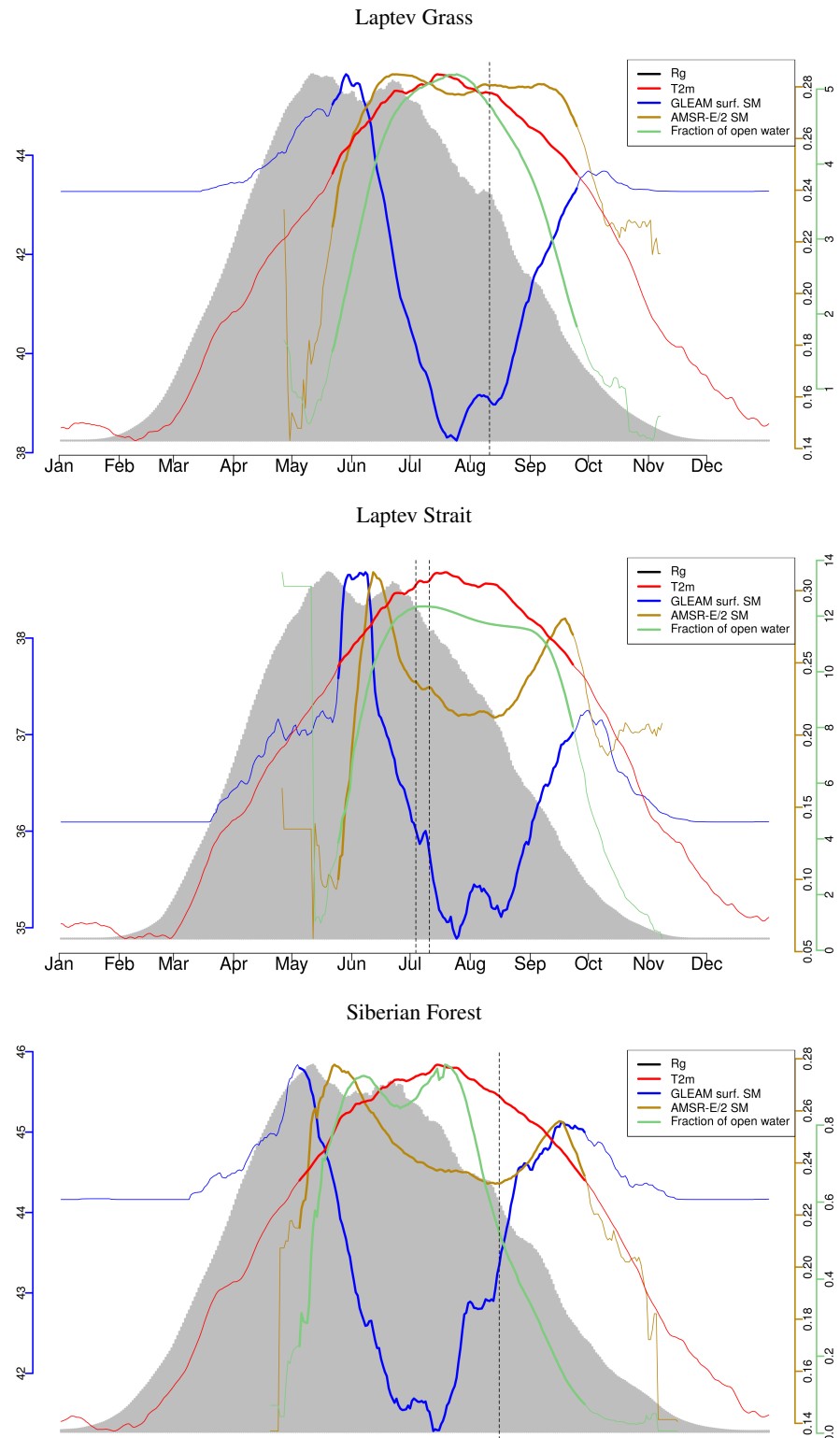

**Figure A2.** Mean seasonal cycle of environmental variables averaged over the areas indicated in Fig. 1.

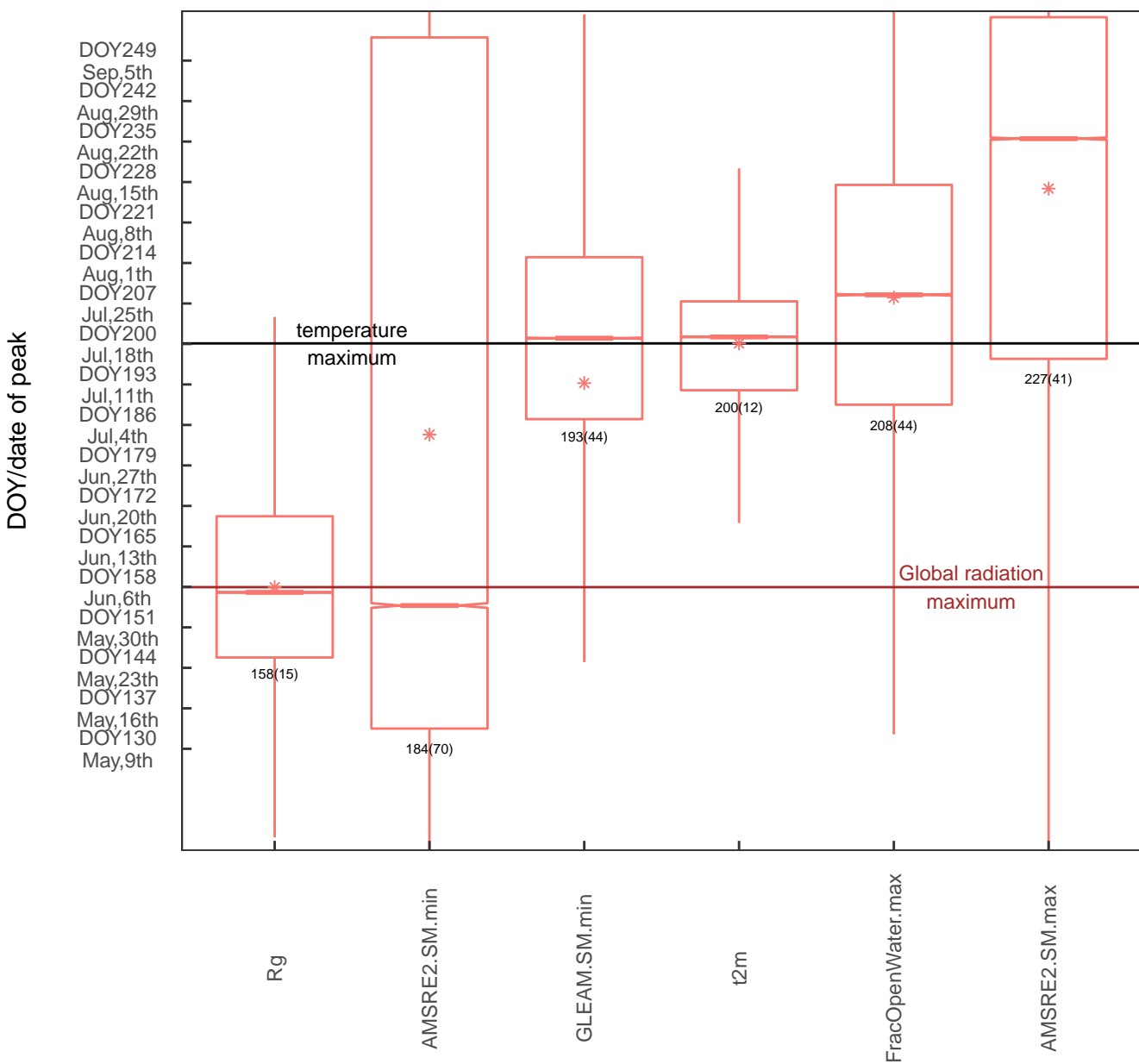

**Figure A3.** Distribution of the DOY of the peak of the different environmental variables over the study region and between years (spatial sampling matched between data sets for each year). Bars in the boxes indicate the median, stars the mean, the numbers below the bars denote the spatial mean (standard deviation).

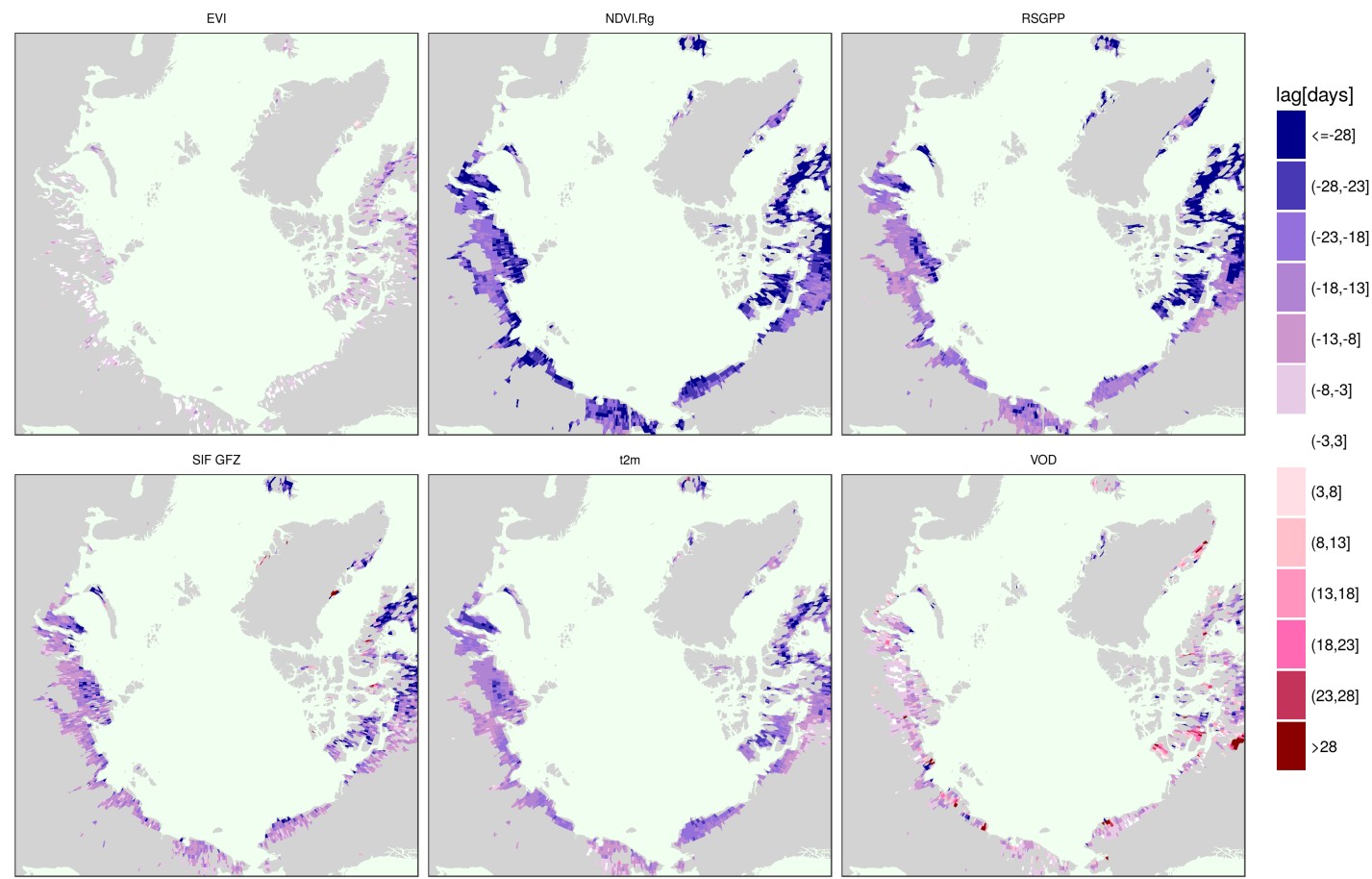

**Figure A4.** Average time difference of the yearly maximum across years between selected vegetation proxies and the NDVI based on 50 bootstrap samples of each year's time series. Shown are only statistically significant differences at a level of 5%.

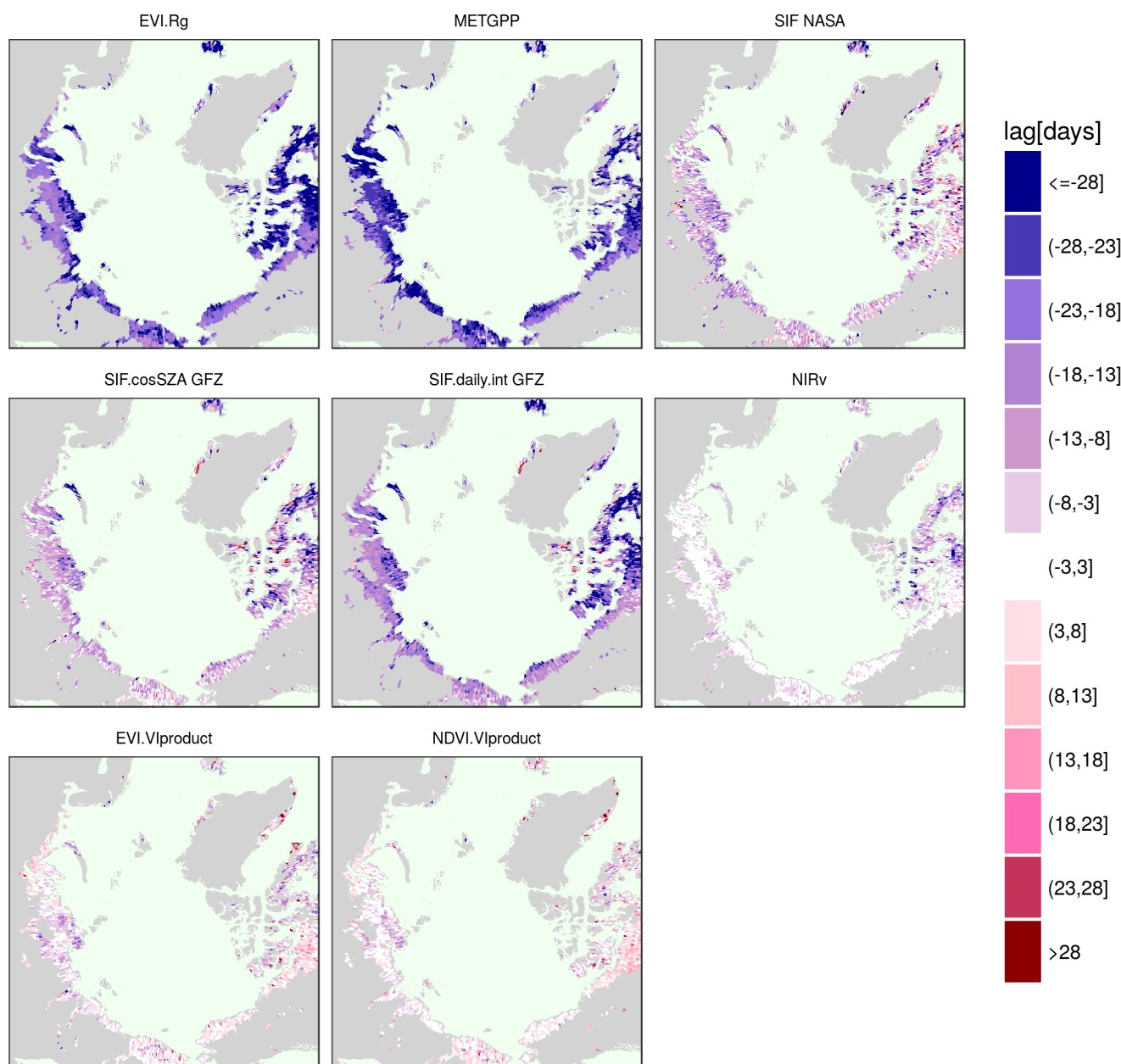

**Figure A5.** Average time difference of the maximum across years of selected vegetation proxies and the NDVI.

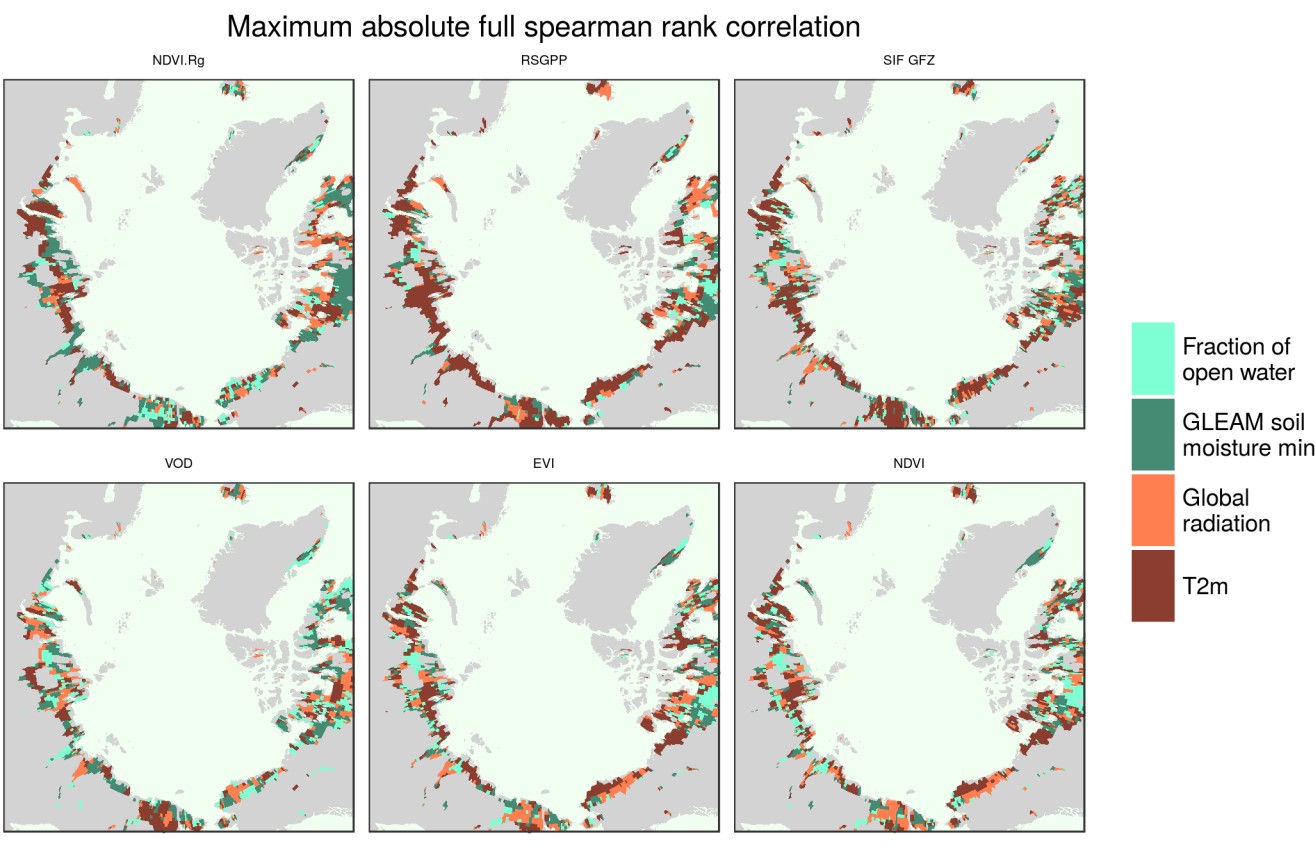

**Figure A6.** Spearman rank correlation between the peak DOY of the vegetation proxies across years and the peak DOY of environmental variables in spatial moving windows of $1.5°$ (so 9 spatial pixels times 10 years at most, correlated only if more than 20 data points available). Plotted here is the variable with the highest absolute correlation without filter for statistical significance. Full correlations have been calculated (no partial correlations). The corresponding spatial distribution of the maximum correlations is given in Fig. A6.

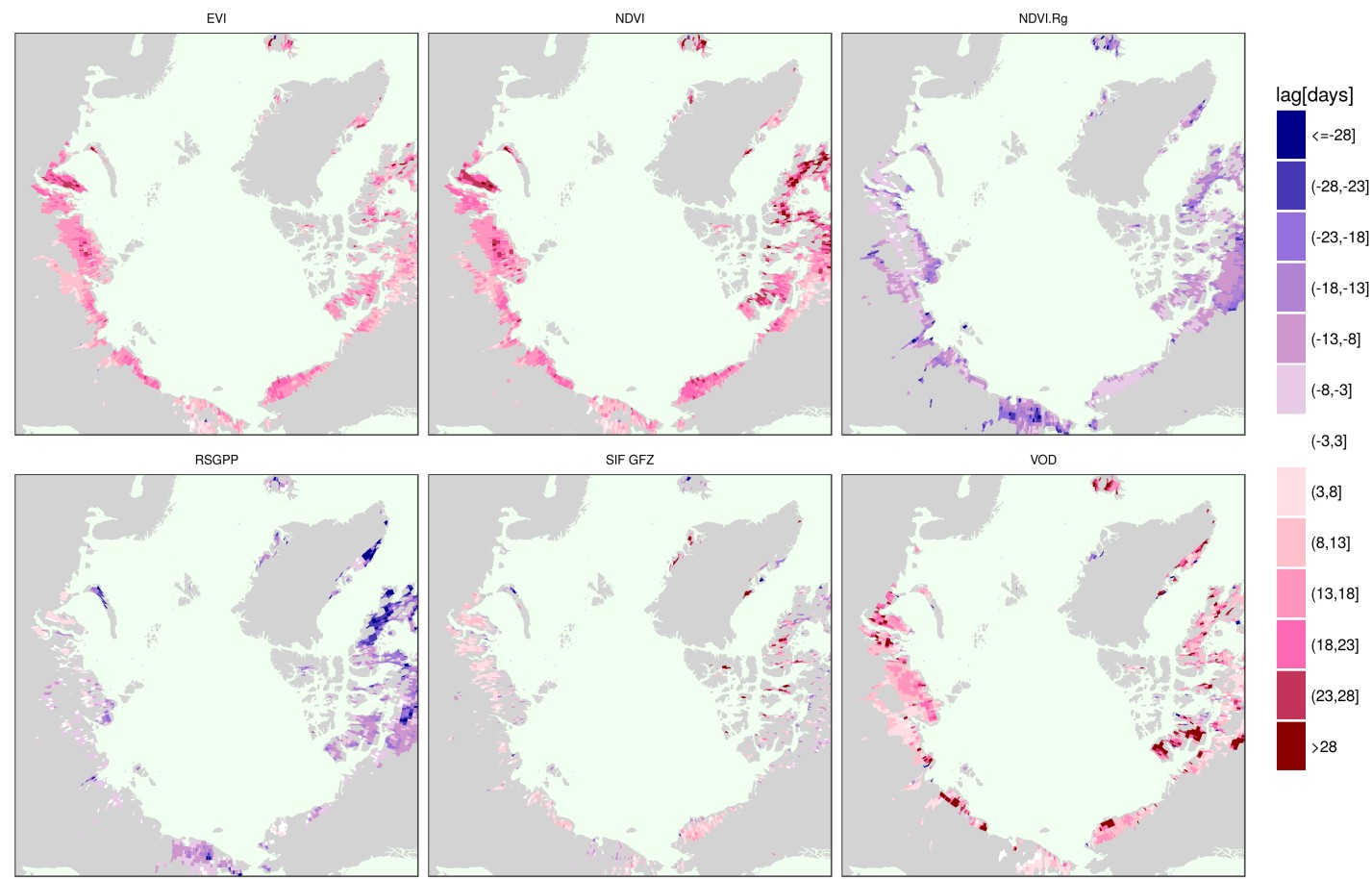

**Figure A7.** Average time difference of the yearly maximum across years between selected vegetation proxies and the air temperature based on 50 bootstrap samples of the time series. Shown are only statistically significant differences at a level of 5%.

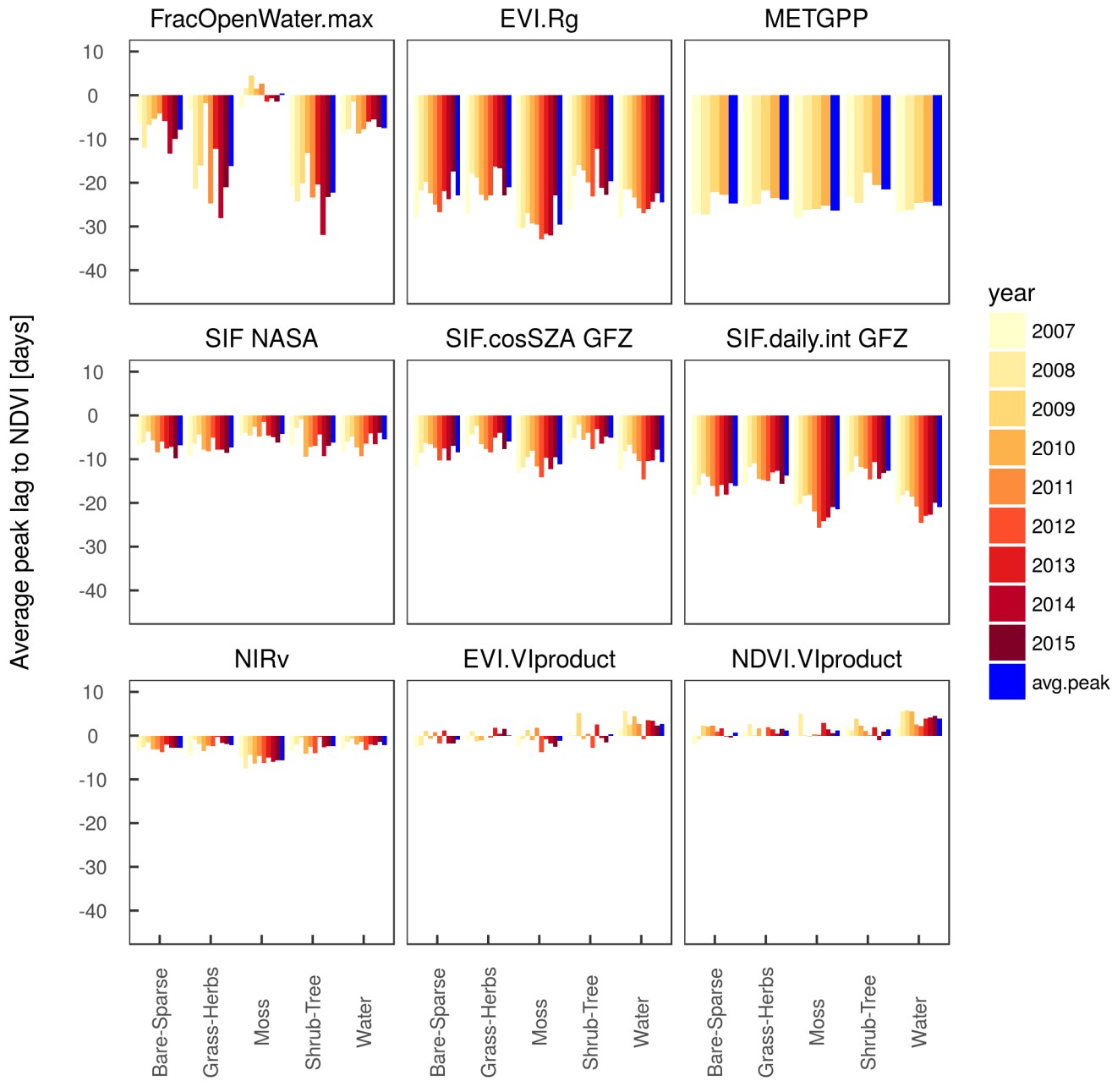

**Figure A8.** Average of the time difference between the peaks of various variables and the NDVI as a reference per vegetation type (based on ESA CCI) and per year. 'avg.peak' represents the average of the lags in the individual years.

| Site-ID | Years | IGBP | Lon | Lat | doi |
|---|---|---|---|---|---|
| NO-Adv | 2012-2014 | WET | 15.923 | 78.186 | 10.18140/FLX/1440241 |
| DK-ZaF | 2008-2011 | WET | -20.5545 | 74.4814 | 10.18140/FLX/1440223 |
| DK-ZaH | 2007-2014 | GRA | -20.5503 | 74.4732 | 10.18140/FLX/1440224 |
| RU-Sam | 2007-2014 | GRA | 126.4958 | 72.3738 | 10.18140/FLX/1440185 |
| RU-Tks | 2010-2014 | GRA | 128.8878 | 71.5943 | 10.18140/FLX/1440244 |
| RU-Cok | 2007-2013 | OSH | 147.4943 | 70.8291 | 10.18140/FLX/1440182 |
| US-Atq | 2007-2008 | WET | -157.4089 | 70.4696 | 10.18140/FLX/1440067 |
| SE-St1 | 2012-2014 | WET | 19.0503 | 68.3541 | 10.18140/FLX/1440187 |
| FI-Lom | 2007-2009 | WET | 24.2092 | 67.9972 | 10.18140/FLX/1440228 |
| DK-NuF | 2008-2014 | WET | -51.3861 | 64.1308 | 10.18140/FLX/1440222 |
| US-Ivo | 2007 | WET | -155.7503 | 68.4865 | 10.18140/FLX/1440073 |
| RU-Ch2 | 2013-2016 | WET | 161.3509 | 68.61689 | 10.18140/FLX/1440181 |

**Table A1.** Sites and years used for a comparison of eddy-covariance derived GPP, SIF footprints within 30 km radius of the tower as well as model GPP from the closest $0.5^circ$ grid cell.

Averaged over all sites and years (Fig. A9) site-GPP peaks about 10 days earlier than site-level air temperature, with effects of up to five days of both the partitioning and the quality filter. Time lags of very similar magnitude are shown by grid cell model GPP. In this respect the patterns at site-level agree with the slightly earlier model GPP peak compared to the temperature peak seen in the regional and interannual analysis (Fig. 4). Conversely, while in the large-scale pattern SIF GFZ and SIF NASA

5   peak close to, but slightly later than the air temperature, this is actually not the case at site-level, where SIF GFZ indicates a maximum at a similar time like model GPP and EC GPP (excluding spurious results from sites NO-Adv, DK-ZaF, DK-ZaH where strong differences between the partitionings are apparent and/or an annual cycle in the SIF data absent, Fig. A12) and *before* air temperature.

For individual sites the patterns are less clear. Despite a frequent tendency to earlier peak EC-GPP, the signs of the time lags

10   between peak EC-GPP and temperature are not consistent. For NO-Adv, RU-Cok, RU-Sam, SE-St1 partly strong dependencies of the peak DOY on the partitioning method and the quality filter are seen. Model GPP is maximized rather earlier than site-level temperature, at some sites the lags become very small. For the sites where the SIF time series show a growing season and not pure noise (this excludes, DK-ZaF, DK-ZaH (Fig. A12), NO-Adv), SIF GFZ reaches its maximum on average earlier than site-level temperature, although there is a large spread of the magnitude of the lag of three to 26 days between sites.

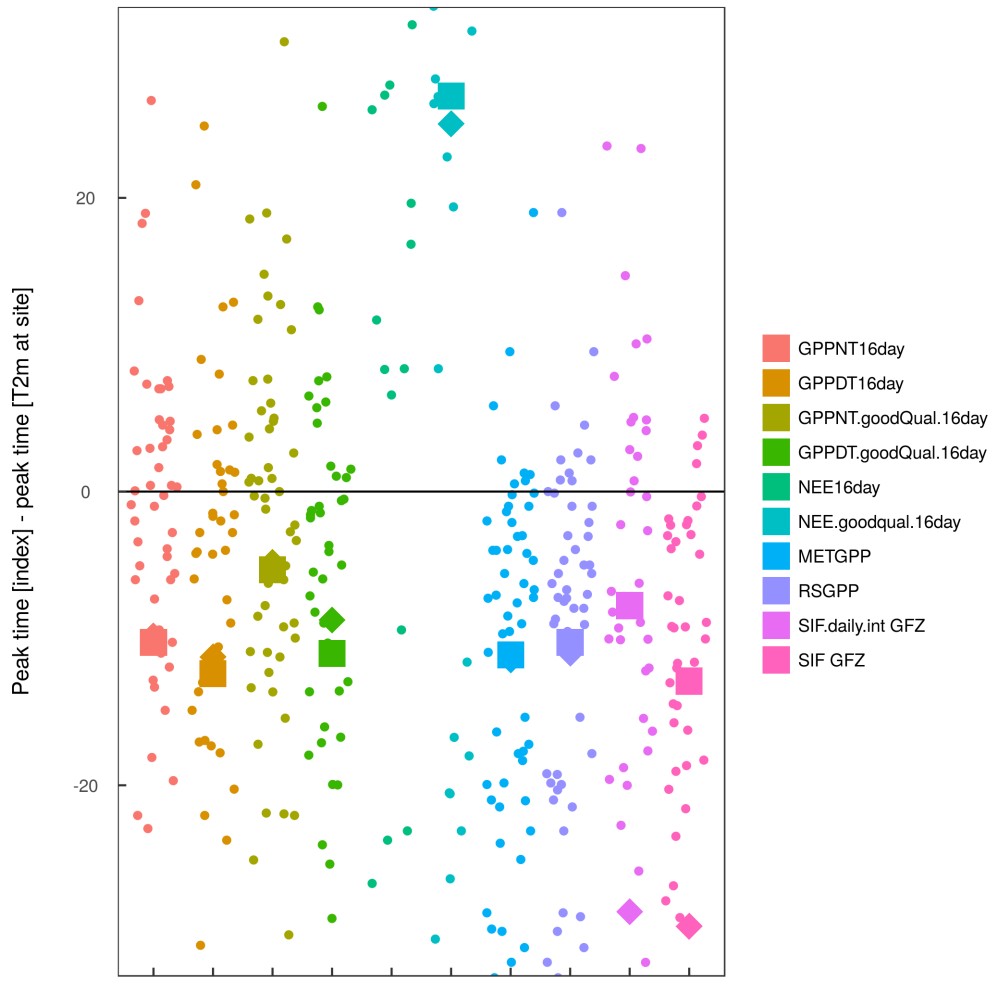

**Figure A9.** Difference in peak timing of the different indicators to site-level temperature over years and at selected FLUXNET sites. Diamonds represent the average peak lag across all years and sites, while squares indicate the average peak time across years and sites but excluding NO-Adv, DK-ZaH,DK-ZaF (where either no seasonality in SIF or strong issues between different partitioning methods are apparent).

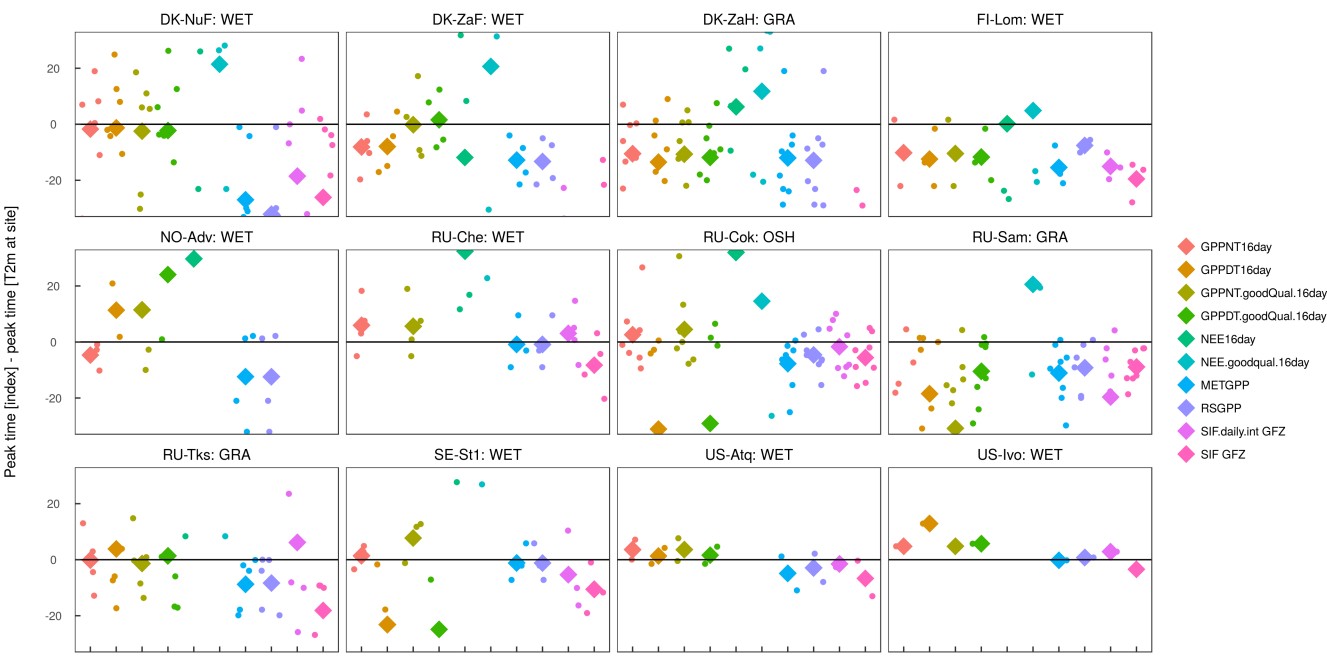

**Figure A10.** Difference in peak timing of the different indicators to site-level temperature over years at selected FLUXNET sites. Diamonds represent the average peak lag across all years per site and variable.

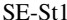

**Figure A11.** SE-St1: Time series (16 day averages) of quality filtered air temperature, NEE, blue bars indicate fraction of available good quality temperature data, gray bars indicate fraction of good and gap-filled NEE data. The middle panel depicts GPP derived from nighttime and daytime partitioning of NEE measurements, filtered for 80% of good quality and gap-filled NEE and also unfiltered. Model GPP is taken from the $0.5^circ$ grid cell with centre coordinates closest to the tower. The bottom panel displays the 16day averages of SIF GFZ footprints within 30 km of the site.

DK-ZaH

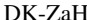

**Figure A12.** DK-ZaH: Time series (16 day averages) of quality filtered air temperature, NEE, blue bars indicate fraction of available good quality temperature data, gray bars indicate fraction of good and gap-filled NEE data. The middle panel depicts GPP derived from nighttime and daytime partitioning of NEE measurements, filtered for 80% of good quality and gap-filled NEE and also unfiltered. Model GPP is taken from the $0.5^{c}irc$ grid cell with centre coordinates closest to the tower. The bottom panel displays the 16day averages of SIF GFZ footprints within 30 km of the site.

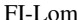

**Figure A13.** FI-Lom: Time series (16 day averages) of quality filtered air temperature, NEE, blue bars indicate fraction of available good quality temperature data, gray bars indicate fraction of good and gap-filled NEE data. The middle panel depicts GPP derived from nighttime and daytime partitioning of NEE measurements, filtered for 80% of good quality and gap-filled NEE and also unfiltered. Model GPP is taken from the $0.5^{c}irc$ grid cell with centre coordinates closest to the tower. The bottom panel displays the 16day averages of SIF GFZ footprints within 30 km of the site.

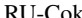

RU-Cok

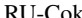

**Figure A14.** RU-Cok: Time series (16 day averages) of quality filtered air temperature, NEE, blue bars indicate fraction of available good quality temperature data, gray bars indicate fraction of good and gap-filled NEE data. The middle panel depicts GPP derived from nighttime and daytime partitioning of NEE measurements, filtered for 80% of good quality and gap-filled NEE and also unfiltered. Model GPP is taken from the $0.5^{c}irc$ grid cell with centre coordinates closest to the tower. The bottom panel displays the 16day averages of SIF GFZ footprints within 30 km of the site.

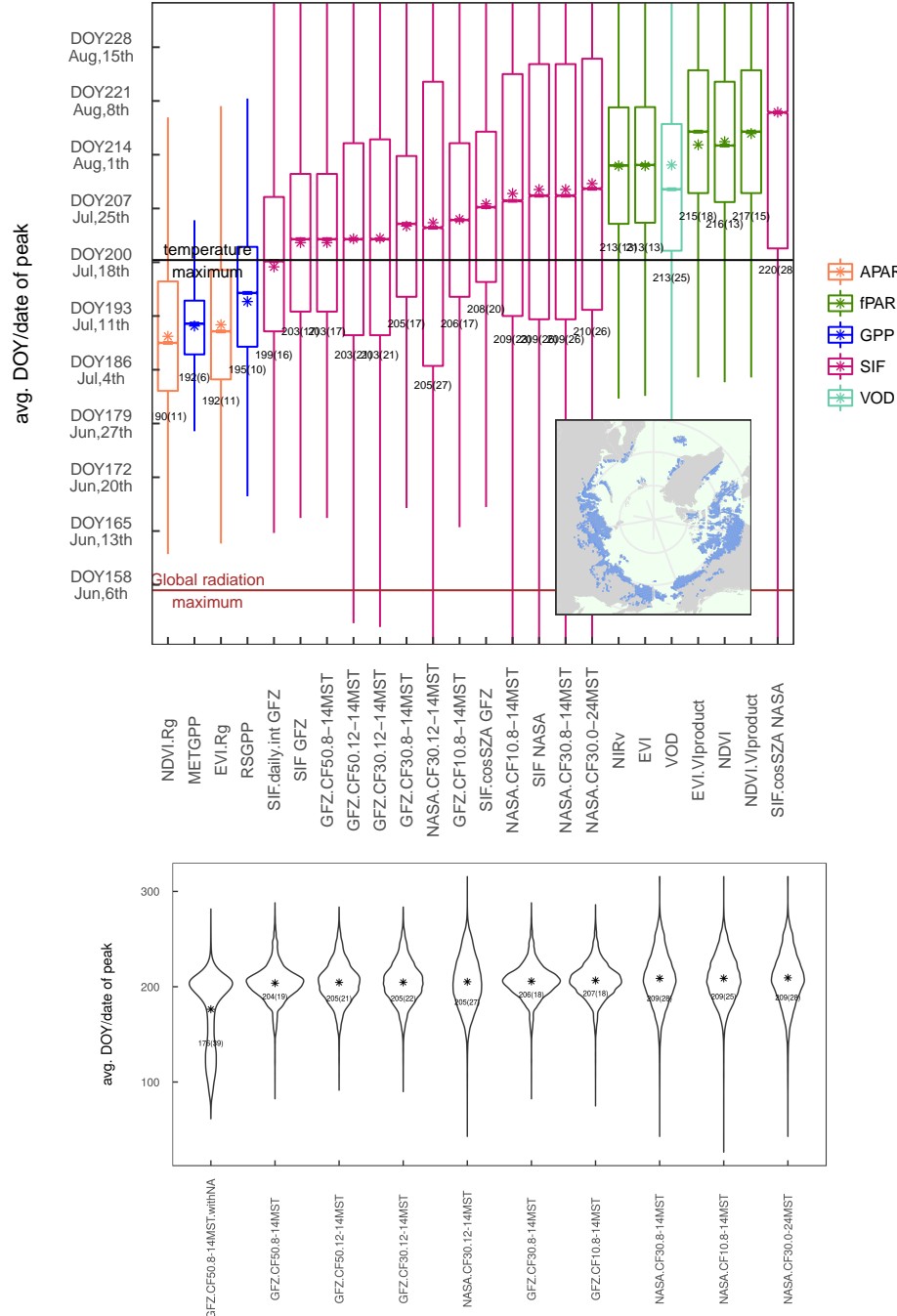

**Figure A15.** Distribution of the annual peak DOY across years and the study area of SIF GFZ and SIF NASA with different filters for cloud cover and overpass time applied. Please note that the sampling of peak values is matched for each year between data sets, which explains slightly differing mean and sd values depending on which vegetation proxies are included in the comparison.

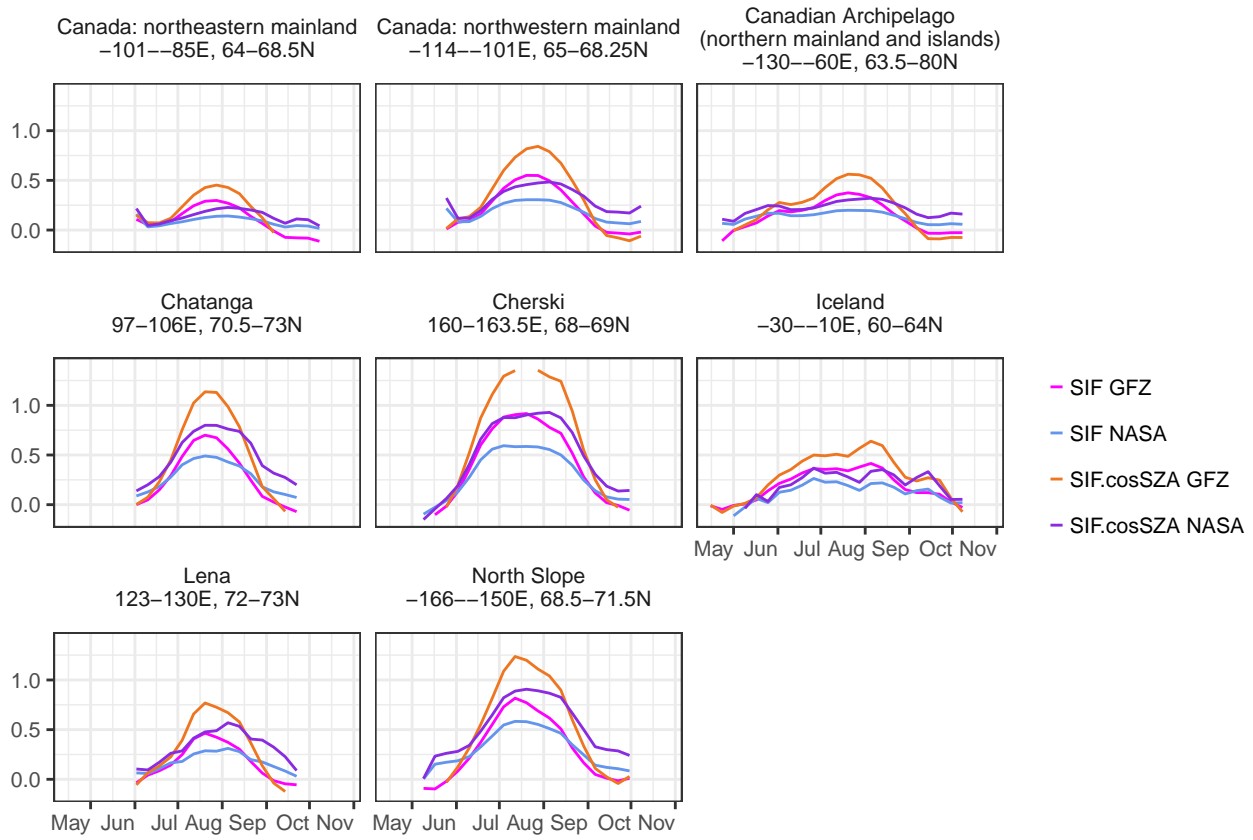

**Figure A16.** Mean seasonal cycles of SIF data from GOME-2 for the two retrievals GFZ and NASA. Values are given in mW/(m2 sr nm).

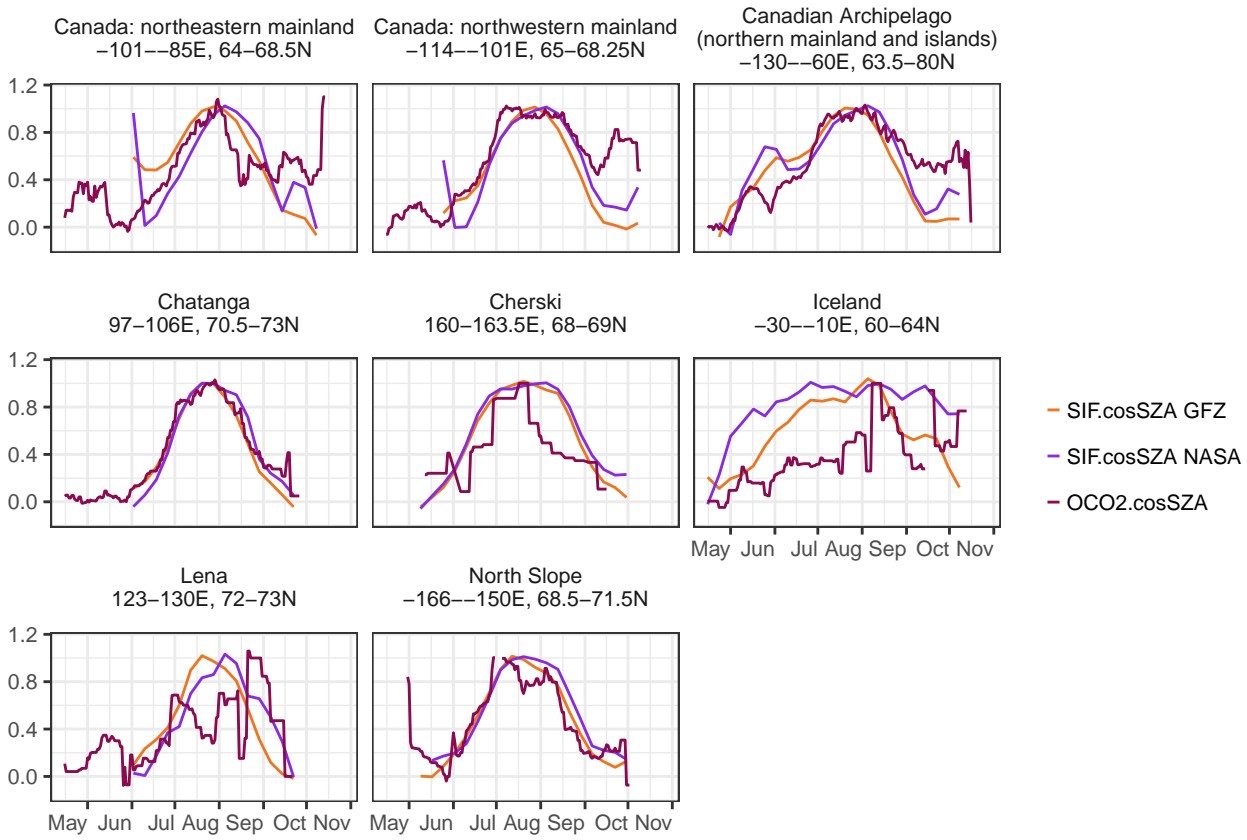

**Figure A17.** Mean seasonal cycles of SIF data from GOME-2 and OCO-2 as a comparison. Values are scaled to 0/1.

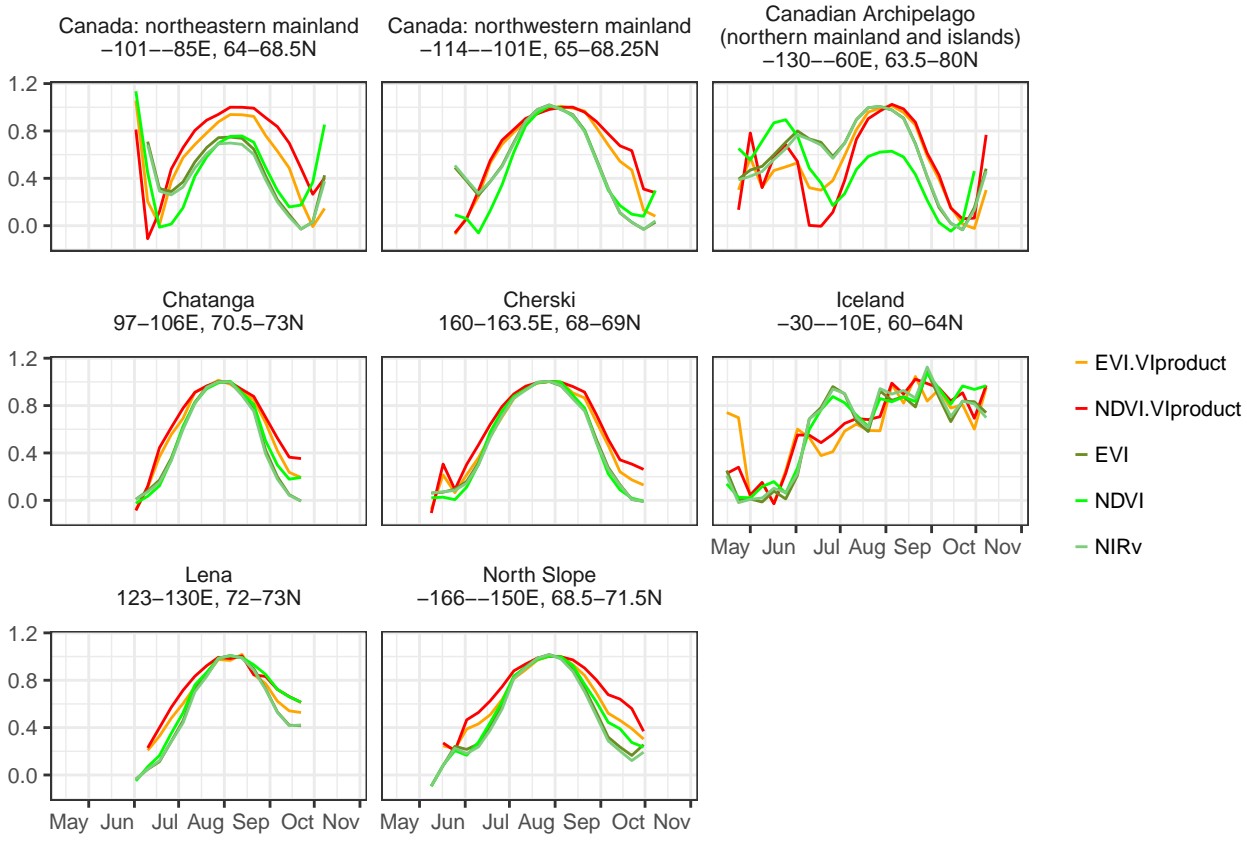

**Figure A18.** Mean seasonal cycles of MODIS vegetation indices calculated from NBAR reflectances (MCD43C4) and as provided by the MODIS VIproduct (MxD13C1). Values are scaled to 0/1.

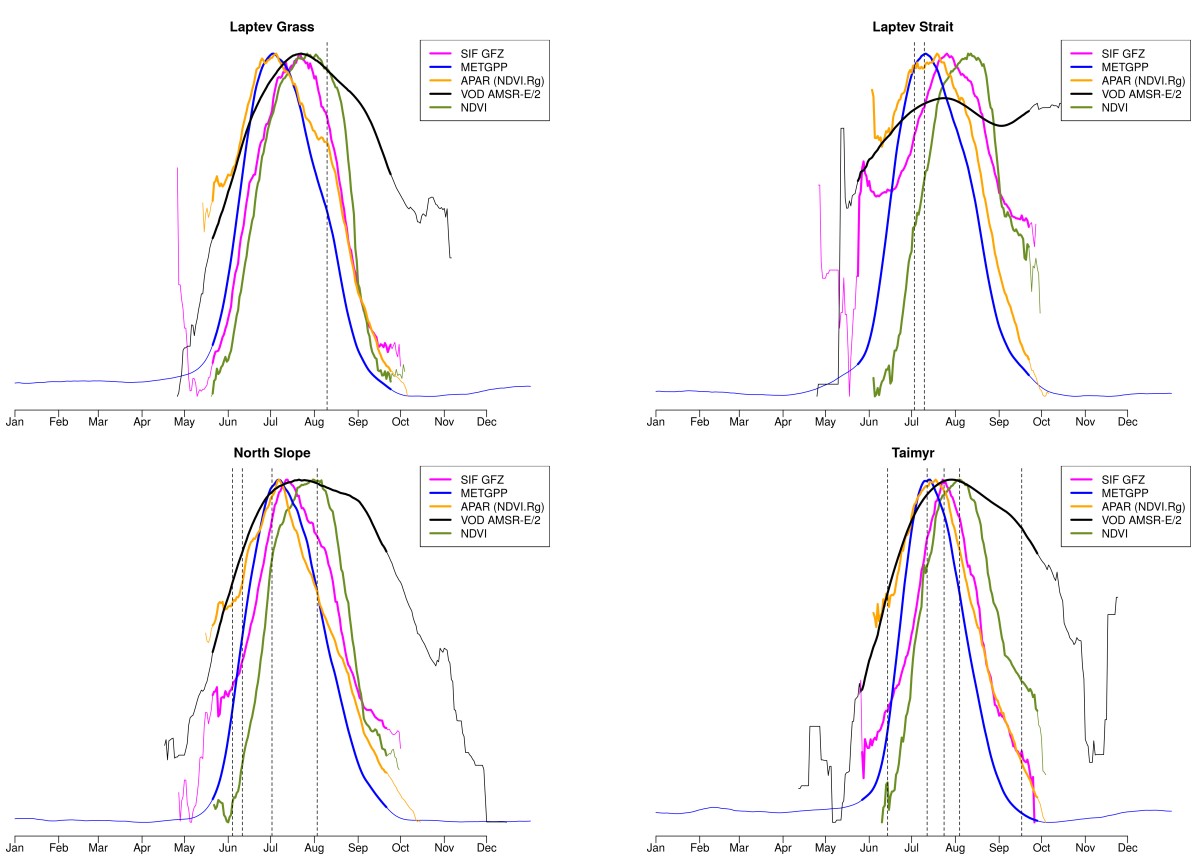

**Figure A19.** Mean seasonal cycles of selected proxies averaged over smaller regions indicated in Fig. 1.