# Peer review of "Assessing the dynamics of vegetation productivity in circumpolar regions with different satellite indicators of greenness and photosynthesis"

_Biogeosciences, 2018_

## Referee Comment (RC1) · Anonymous Referee #1 · 23 May 2018

Summary and Recommendation

This paper "Assessing the dynamics of vegetation productivity in circumpolar regions with different satellite indicators of greenness and photosynthesis" by Sophia Walther and co-authors provides an empirical analysis of the timing of peak vegetation productivity across pan-Arctic tundra as observed from satellite observed greenness and photosynthesis. The authors recognize and attempt to account for the many issues in observing this remote environment from space, combining multiple complimentary vegetation indicators/proxies and applying careful quality control. Analysis of environmental variables including temperature, soil moisture, and open water fraction provides

key mechanistic context. The introduction of background material is thorough and well written, and sets the stage well.

Their results indicate a pattern of differential phasing of annual peaks of vegetation proxies, starting with APAR, GPP, SIF, and Vis/VOD. This pattern is clear in the spatial and temporal mean, and although subject to significant uncertainty (which is quantified in figures but not in the main text), verifies some a priori expectations, such as a delay in SIF/GPP onset relative to APAR consistent with early season NPQ, but with an interesting surprise in the ordering of GPP before SIF, which may or may not be realistic and warrants further investigation. The use of VOD is novel, but the analysis is brief and doesn't add much to our phenological understanding of tundra ecosystems. More analysis and discussion is strongly encouraged; in particular, the authors missed an opportunity to discuss the potential use of VOD in this and/or future studies to explain seasonal to interannual changes in greenness (from EVI and NDVI) as a function of vegetation water content. I also wonder why the authors didn't assess satellite-based findings against ample flux tower data available across the pan-Arctic over the lengthy study period (2007-2016), for example by comparing FLUXCOM GPP and satellite SIF to tower GPP to confirm or falsify the differential phasing between APAR, GPP, and SIF.

Overall, I recommend this important study be accepted after the points discussed above and in major/minor comments below are addressed.

Major Comments

P7, L30-31: In the early growing season prior to soil thaw, AM SIF is delayed relative to PM SIF, leading to biases in SIF recovery as observed from predominantly morning overpasses. Comparisons to tower data support this (Parazoo et al. 2018). Although unlikely to affect peak SIF timing following thaw, I recommend filtering SIF into AM and PM values to test seasonal cycle phasing as a function of time of day.

Parazoo, N. C., Arneth, A., Pugh, T. A., Smith, B., Steiner, N., Luus, K., ... & Rödenbeck, C. (2018). Spring photosynthetic onset and net CO 2 uptake in Alaska triggered

by landscape thawing. Global change biology.

P8, L4-5: Peak SIF timing differences between NASA and GFZ datasets could be related to different cloud screening criteria (clf = 0.3 for NASA and 0.5 for GFZ). Please test GFZ seasonality for clf = 0.3.

P8, L34: It's not clear if readers should treat FLUXCOM GPP differently from SIF in the interpretation of peak photosynthetic activity. Do the authors intend here to have multiple measures of photosynthesis, or is the purpose for SIF to provide a dynamical interpretation of vegetation proxies?

P10, L9: GLEAM SM looks very different from AMSR-E/2 (blue line in Fig. 2 & 3). The blue line isn't mentioned in the captions or in environmental analysis at the end of Section 2.8 (P10, L1-12) so it's difficult to tell if the differences a plotting error or if they are physical. Please explain. In general, AMSR-E/2 is only referred to one time in the results and discussion, and thus the authors might consider dropping this dataset.

P12, L5-7: I'm not clear what is being done here and why. It appears that for each satellite proxy, the timing of its annual maximum is correlated with the timing of 4 environmental variables for each year (2007-2016?), and the variable with the highest correlation is selected and color coded spatially. Is this correct? Is screening for statistical significance applied? It seems the point here is more to determine if there is a good predictor for peak vegetation activity, rather than prove that environmental-vegetation dynamics hold across years. Please clarify and explain.

P12, L7-10: As a follow up, the above relationships are difficult to pick out in figure 7, although I admit they do appear to emerge as a squint my eyes. Perhaps this can be better quantified in a histogram (or table) of fractional areal coverage for each variable (e.g., what percent of land is NDVI.Rg best correlated to (a) fraction of open water, (b) soil moisture, etc.)

P12, Section 3.4: There is no mention of VOD. It is highly variable between years and

across land covers. Please discuss. Also, what's going on in 2016 in 't2m'? Soil moisture, fraction open water, and VOD are also anomalous in 2016. Are these correlated? Please discuss

P12, L 26: I wouldn't classify these relationships as clear, due mainly to excessive variability in mean date and lag time. Certainly, this study has illuminated some interesting and potentially important emergent properties which warrant further investigation.

P12, L28: Please report uncertainties on these numbers, as these patterns are not statistically significant

P12, L32: VOD analysis needs more elaboration throughout the MS. In general, the study finds similar behavior to EVI and NDVI. Perhaps this means VOD doesn't tell us anything we don't already know from EVI and NDVI? Or rather, this suggests that seasonal and interannual changes in greenness are driven by vegetation water content, and potentially could help explain longer term change associated with greening and browning (drying vs wetting of vegetation). Please discuss.

P13, L20-22: This is the first mention of (prognostic) models, and it is rather brief and confusing. Are the authors saying that Earth system models have is wrong, due to reversed seasonal dependencies on LAI, light, and temperature? This is important insight and worth more discussion.

P13, L24-34:

a. Mean time lag is 6 to 11 days depending on SIF product.

b. It would be helpful if the authors could plot a seasonal time series of APAR, GPP, and SIF with arrows indicating how changing light absorption affects the timing of peak GPP (lower light & temp) and SIF (higher light & temp), assuming the differential phasing is physical.

c. The explanation of the difference between APAR and SIF is explained, but it's not clear from the discussion why instant SIF and SIF.cos(SZA) peak after GPP. The paragraph discusses the theory that light saturating conditions leads to correlation of fluorescence and photosynthesis, suggesting the peaks are synchronized, not offset. Parazoo et al. (2018) show synchrony of SIF, SIF driven GPP, and tower GPP in summer, but an offset in spring, suggesting an error in GPP data. However it's possible the SIF delay reflects conditions of diverging SIF and GPP under increased heat and water stress later in summer, with time of day again playing an important role in observed phasing. Thus, it is worth discussing all of the range of outcomes that the SIF delay is (a) physical, (b) related to error in model GPP as discussed later (P15, L22-34), and (c) negligible due to high uncertainty and variability in all datasets.

Minor Comments

P9, L27-29: 2009 typo? Somewhere in this line the authors appear to extrapolate from 2012 to 2009, but it's not specifically mentioned

P12, L11: "less frequently" is vague. Please quantify.

P12, L20: "shrubs and trees" -> more precisely, "mixed shrub-tree land covers"

P16, L4: "second third of July" is awkward. Might rephrase as "one week later"

P16, L12: might check out parallel study by Parazoo et al (2018)

---

## Referee Comment (RC2) · Anonymous Referee #2 · 29 May 2018

Summary: This paper analyzes the timing of peak vegetation productivity for high latitude treeless ecosystems based on spaceborne remote sensing observations. Satellite data utilized in the study includes: 1) MODIS-based GPP; 2) MODIS NDVI; 2) MODIS-based APAR; 3) GOME2 SIF; and 4) AMSR VOD. The authors find a consistent ordering of peaks: APAR < GPP < SIF < VIs / VOD. The authors conclude that the consistent differences between photosynthetic activity and greenness is an important consideration when using satellite observations as drivers in vegetation models. Overall, I find this to be a nice paper with some interesting/useful findings. I do however have a number of recommended revisions before I can recommend this paper for publication. Most importantly, I highlight a number of ways to potentially improve the

analysis and better demonstrate robustness of the findings.

Major Comments: 1. The introduction is very well written and does an excellent job of framing the research questions and establishing the importance of the work. 2. The processing of the various satellite observations is explained very well and done appropriately. 3. The authors use a definition of the timing of peak vegetation productivity as "the timing of the annual maximum is defined as the average day of year (DOY) of all days at which the values exceed the 95th quantile of all valid values of the time series in a year in a given pixel." I question this method as these different data have inherently different levels of daily variability, which could result in spurious dates being included in the period exceeding the 95th quantile of all valid values. I recommend filtering (e.g., using a Savitzky-Golay filter) and then finding the peak of the time series. This would be a good check on the robustness of the main findings of the paper. 4. Additionally, I find the examination of the peak of seasonal activity interesting, but feel it would be complemented by consideration of the start and end of season. Using the above-mentioned Savitzky-Golay filter (or something similar), start and end of season estimates could be easily derived. I recognize this is a lot of additional work and analysis, but I think it could really strengthen the findings of the paper. I put this forward as a recommendation that could strengthen the paper, but this additional analysis is not necessary needed to warrant publication as the current findings of the paper are still very interesting. 5. I think the study would benefit from comparison at any eddy covariance flux tower sites within the study area. EC flux tower-derived GPP data could be used more effectively than modeled GPP to establish which proxies are capturing peak photosynthesis and which are capturing other aspects of plant dynamics (e.g., changes in water content, changes in leaf area, etc.). In particular, this would be useful in establishing whether SIF observations are providing new information more directly associated with plant physiological function. For instance, the authors may find SIF better matches EC tower-based GPP compared to the modeled GPP used in the analysis. This would be very useful information for the modeling community. 6. Page 16, line 15: "Furthermore, the fact that the SIF maximum is reached in close temporal agreement

with air temperature indicates a benefit for photosynthesis from highest temperatures." Without comparing to actual GPP observations (e.g., from EC flux towers), I do not think this is a valid conclusion. There is nothing in this analysis to conclusively show SIF is actually tracking GPP since it is only compared to modeled GPP.

Minor Comments: Page 7, line 15: time should be corrected to "1:30 AM". Page 8, Line 14: grammar error: "and it using such scaling factor may further amplify noise."

---

## Author Comment (AC1) · 10 Jul 2018

Summary and Recommendation This paper "Assessing the dynamics of vegetation productivity in circumpolar regions with different satellite indicators of greenness and photosynthesis" by Sophia Walther and co-authors provides an empirical analysis of the timing of peak vegetation pro- ductivity across pan-Arctic tundra as observed from satellite observed greenness and photosynthesis. The authors recognize and attempt to account for the many issues in observing this remote environment from space, com- bining multiple complimentary vegetation indicators/proxies and applying careful qual- ity control. Analysis of environ- mental variables including temperature, soil moisture,

and open water fraction provides key mechanistic context. The introduction of background material is thorough and well written, and sets the stage well. Their results indicate a pattern of differential phasing of annual peaks of vegetation proxies, starting with APAR, GPP, SIF, and Vis/VOD. This pattern is clear in the spatial and temporal mean, and although subject to significant uncertainty (which is quantified in figures but not in the main text), verifies some a priori expectations, such as a delay in SIF/GPP onset relative to APAR consistent with early season NPQ, but with an interesting surprise in the ordering of GPP before SIF, which may or may not be realistic and warrants further investigation. The use of VOD is novel, but the analysis is brief and doesn't add much to our phenological understanding of tundra ecosystems. More analysis and discussion is strongly encouraged; in particular, the authors missed an opportunity to discuss the potential use of VOD in this and/or future studies to explain seasonal to interannual changes in greenness (from EVI and NDVI) as a function of vegetation water content. I also wonder why the authors didn't assess satellite-based findings against ample flux tower data available across the pan-Arctic over the lengthy study period (2007-2016), for example by comparing FLUXCOM GPP and satellite SIF to tower GPP to confirm or falsify the differential phasing between APAR, GPP, and SIF. Overall, I recommend this important study be accepted after the points discussed above and in major/minor comments below are addressed.

++We agree with your comments and followed your suggestions in that we quantified uncertainties in the main text (e.g. in the evaluation of Fig.4), included a comparison of model GPP and SIF to EC towers in an additional plot (as supporting information as the results are not conclusive). We also extended the discussion on the peak timing of the VOD.

Major Comments P7, L30-31: In the early growing season prior to soil thaw, AM SIF is delayed relative to PM SIF, leading to biases in SIF recovery as observed from predominantly morning overpasses. Comparisons to tower data support this (Parazoo et al. 2018). Although unlikely to affect peak SIF timing following thaw, I recommend filtering
SIF into AM and PM values to test seasonal cycle phasing as a function of time of day. Parazoo, N. C., Arneth, A., Pugh, T. A., Smith, B., Steiner, N., Luus, K., ...& Rödenbeck, C. (2018). Spring photosynthetic onset and net CO 2 uptake in Alaska triggered by landscape thawing. Global change biology.

++We agree that both overpass time and cloud fractions might affect the retrieved timing of the annual peak. We therefore tested this by creating from both the GFZ and NASA L2 data different data sets with different combinations of filters for cloud fractions and overpass times applied. Indeed, a stricter cloud filter of 0.3 instead of 0.5 delays the SIF GFZ peak slightly, bringing it into closer but not full temporal agreement with the SIF NASA peak. Strict filtering for midday over pass times advances (as expected) the annual peak, but for other combinations of overpass filters no systematics appear. We conclude that the different cloud fractions can explain a part of the difference between the SIF GFZ and SIF NASA and argue that the remainder can be ascribed to the different absolute values and the effects of noise in the two data sets that might make a reliable peak identification in SIF NASA more difficult than in SIF GFZ and also lead to a larger spread. We included these plots in the supporting information and added discussion on the issues in the manuscript. Please note, that 1) the definition of cloud fraction is not exactly identical between GFZ and NASA data sets (we included the information in the methods section of the revised manuscript and outline them below) and 2) that in the following plots (Fig.1 and Fi.2) the sampling of peak values is matched for each year between data sets, which explains slightly differing mean and sd values depending on which vegetation proxies are included in the comparison.

(Details on how the cloud fractions in SIF NASA and SIF GFZ are obtained: For the cloud fraction retrieval the two data sets use similar but not identical approaches. In the NASA retrieval, cloud radiance fractions at 865nm (Joiner et al. 2012, AMT) are used to infer the effective cloud fraction. When reflectances are smaller than the climatology, negative CF can be obtained and after personal communication with Joanna Joiner they are to be treated as zero and are not filtered from the data set. Generally, in the

NASA v26 data no retrievals with cf>0.3 are provided. All retrieved values irrespective of the measurement time are provided. Conversely, the GFZ data use the FRESCO cloud mask that is provided with GOME2 L1b data (Köhler et al. 2015, AMT) that used reflectances in the O2-A band to obtain effective cloud fractions (Wang et al. , 2008, Atmos. Chem. Phys.). In this case, no negative values occur, but missing values are found in the data set. In the submitted manuscript they were masked out. Here, we also include a version in which we retain the fluorescence values in the time series were the CF is missing. Also, in the L2 GFZ SIF data filters for CF<=0.5 and overpass times of 8-14MST are applied. ###

P8, L4-5: Peak SIF timing differences between NASA and GFZ datasets could be related to different cloud screening criteria (clf = 0.3 for NASA and 0.5 for GFZ). Please test GFZ seasonality for clf = 0.3. ++We did the test and confirm an effect on the peak difference between the two data sets as shown in the answer to the previous comment. ###

P8, L34: It's not clear if readers should treat FLUXCOM GPP differently from SIF in the interpretation of peak photosynthetic activity. Do the authors intend here to have multiple measures of photosynthesis, or is the purpose for SIF to provide a dynamical interpretation of vegetation proxies? ++We agree that the formulation 'A different indicator of photosynthetic activity is provided by the GPP model simulations from the FLUXCOM initiative' can be confusing. Fluxcom GPP is used next to SIF as an indicator of photosynthesis. To clarify we changed in the manuscript the line to 'Another indicator of photosynthetic activity is provided by the GPP model simulations from the FLUXCOM initiative'. ###

P10, L9: GLEAM SM looks very different from AMSR-E/2 (blue line in Fig. 2 & 3). The blue line isn't mentioned in the captions or in environmental analysis at the end of Section 2.8 (P10, L1-12) so it's difficult to tell if the differences a plotting error or if they are physical. Please explain. In general, AMSR-E/2 is only referred to one time in the results and discussion, and thus the authors might consider dropping this

dataset. ++Thanks for pointing this out. Indeed, the very different behaviour of the two data sets cannot be fully explained and for both it is unclear what the meaning in tundra ecosystems is. We still think that this discrepancy is worth being shown and decide to include the full annual cycles of both soil moisture data sets with axes in Fig. 1 & 2 and include a short discussion on possible reasons for the discrepancy in the methods section (Fig.3.&4). 'Directions of possible explanations might be: 1) Both refer to slightly different quantities. AMSR-E/2 denotes the volumetric soil moisture in the uppermost 1cm of land surface while GLEAM refers to surface soil moisture (denoted as 0-10cm depth). Full comparability might thus not be given. 2) The applicability of GLEAM data for high latitudes has not been shown. 3) The fraction of open water from AMSR-E/2 shows surprising behaviour in that we expected the annual maximum to appear in early summer and not rather late in the summer. In the examples above of different behaviour between GLEAM and AMSR-E/2 soil moisture, the latter shows similarly unexpected behaviour. This is indicative of the problems that still need to be faced for soil moisture retrievals in the high latitudes. 4) It is actually not clear what these quantities mean in tundra ecosystems. The microwave signal from the persistently wet peat base of the moss layer can penetrate the dry moss layers (which might reach thicknesses of tens of centimeters), but the signal in GLEAM in similar cases is unclear. '

5) In an earlier version we had used soil moisture data from AMSR-E obtained with another retrieval technique. This data set has apparently been deprecated and we dropped it from the analysis. In this data set we found still different patterns. For us this is a clear indication of the problems that still need to be faced for soil moisture retrievals in the high latitudes. 6) Plotting error. We checked all scripts and could not find an error. ###

P12, L5-7: I'm not clear what is being done here and why. It appears that for each satellite proxy, the timing of its annual maximum is correlated with the timing of 4 environmental variables for each year (2007-2016?), and the variable with the highest correlation is selected and color coded spatially. Is this correct? Is screening for statistical significance applied? It seems the point here is more to determine if there is a good predictor for peak vegetation activity, rather than prove that environmental-vegetation dynamics hold across years. Please clarify and explain. ++Here we asked whether the relationships between the environmental variables and the different vegetation proxies hold across years, e.g. whether an early temperature peak would simultaneously be mirrored in an earlier photosynthesis peak and therefore temperature (or any other environmetnal variable) emerges as the most important driver. For this, we correlated the peak DOYs of each proxy with each environmental variable using all years and a spatial moving window. In Fig.7, the correlations have not been filtered for statistical significance (only few scattered pixels have significant relationships) but are intended to give a general idea of whether the proxies' annual maximum is rather related to moisture or energy related variables. ###

P12, L7-10: As a follow up, the above relationships are difficult to pick out in figure 7, although I admit they do appear to emerge as a squint my eyes. Perhaps this can be better quantified in a histogram (or table) of fractional areal coverage for each variable (e.g., what percent of land is NDVI.Rg best correlated to (a) fraction of open water, (b) soil moisture, etc.) ++Yes, the relationships that we claim to see are admittedly hard to identify from the maps. We therefore decided to move the previous Fig. 7 to the SI and summarize the maps as barplots (new Fig.7, here Fig.5) from which the slightly stronger importance of energy variables for most vegetation proxies emerges more clearly. ###

P12, Section 3.4: There is no mention of VOD. It is highly variable between years and across land covers. Please discuss. Also, what's going on in 2016 in 't2m'? Soil moisture, fraction open water, and VOD are also anomalous in 2016. Are these correlated? Please discuss ++This has partly been a bug, we remove data from 2016 from the analysis. We will further include a more extensive discussion on the variability between years in the manuscript. ###

P12, L 26: I wouldn't classify these relationships as clear, due mainly to excessive variability in mean date and lag time. Certainly, this study has illuminated some interesting and potentially important emergent properties which warrant further investigation. ++We changed the tone of this sentence and now it reads: 'Despite the considerable challenges for remote sensing applications in high latitudes and inherent comparatively large variability in the timing of the annual maximum, the differences in the peak timing of families or groups of key satellite indicators of plant productivity appear to be ordered in polar tundra.' ###

P12, L28: Please report uncertainties on these numbers, as these patterns are not statistically significant ++We have indicated standard deviations of the peak timing across space and years in brackets in Fig.1 and will mention the large spread indicated by them in this first paragraph of the discussion. For a better quantification of uncertainties and the robustness of the results we further plan to use bootstrapping of the time series (resample 50 time series per pixel, year and vegetation proxy with consistent sampling between proxies) and will include the results in the revised manuscript. ###

P12, L32: VOD analysis needs more elaboration throughout the MS. In general, the study finds similar behavior to EVI and NDVI. Perhaps this means VOD doesn't tell us anything we don't already know from EVI and NDVI? Or rather, this suggests that seasonal and interannual changes in greenness are driven by vegetation water content, and potentially could help explain longer term change associated with greening and browning (drying vs wetting of vegetation). Please discuss. ++The discussion of the VOD results was indeed very short in the previous version. Following your suggestion we included an additional paragraph in the discussion on the similar timing between VOD and vegetation indices. We do, however, not feel confident enough to overinterpret the VOD results as it is not straightforward what VOD in the specific context of tundra actually means. The additional paragraph reads like this: 'The highest vegetation water content is indicated by the VOD at a very similar time like the peak values of the vegetation indices - which might corroborate the usefulness of VOD to indicate

vegetation biomass also in tundra ecosystems. Both vegetation indices and VOD are sensitive to vegetation structure and density, VOD in addition to water content (Liuetal2011). Especially in low biomass regions - as applied for tundra - a linear relationship between VOD and vegetation water content has been found (Teubneretal2018). The similarity of VOD as an indicator of total aboveground biomass to the vegetation indices might also support our interpretation of delayed peak greenness compared to photosynthesis due to longer lasting build-up of plant material and pigments and indicate that the proposed relationship by Teubneretal2018 of VOD-vegetation water content-LAI to GPP does not fully hold in tundra. Conversely, when looking at examples of the mean seasonality of VOD in comparison to the other vegetation proxies for selected regions (new SI Figure, see below Fig.6-9) the VOD annual cycle appears broader and the peak less well defined than the one of the vegetation indices. This could indicate that the vegetation water content changes only slightly during the growing season while possibly the chlorophyll concentrations independently exhibit more pronounced dynamics and affect the vegetation indices more strongly. Another possible factor contributing to the broader annual cycle is VOD being related to the water content of the total aboveground biomass, including moss and woody components and litter. If persistently wet, moss might drive the VOD signal but less strongly the greenness signal. There is a high sensitivity of moss to air humidity as a consequence of the absence of roots. Despite a wet peat layer there might be several centimetres of dry moss material. In grass lands, consistently high emissions and consequently a lower seasonality were ascribed to the contributions of litter and wet vegetation components (Grantetal2016). Similar mechanisms might also hold for the VOD observations in tundra, especially considering that carbon turnover is slow in the high latitudes. It is also not clear what the VOD signal over water saturated soil might be. Moreover, the retrieval of VOD is strongly dependent on the representation of open water and soil moisture. Considering that the retrieval algorithms have not been calibrated for tundra-like conditions and that with the high heterogeneity regarding plant types and landscape components it might be difficult to accurately separate the contributions of vegetation from soil and water, it

is not clear what VOD in tundra means. In future studies it would be useful the analyse a suite of different VOD products from different sensors and wavelengths together with in-situ observations in order to understand whether greenness and vegetation water content are strongly coupled in tundra or not and to what extent different retrievals affect the result. ' ###

P13, L20-22: This is the first mention of (prognostic) models, and it is rather brief and confusing. Are the authors saying that Earth system models have is wrong, due to reversed seasonal dependencies on LAI, light, and temperature? This is important insight and worth more discussion. ++The reference to the land surface models is brought up in order to give support for our interpretation of the ordering of the peaks of the different proxies in that temperature is always a constraint for GPP, but LAI only before the peak of the growing season – consistent with light saturation. The reference also states that after the GPP maximum light is the limiting factor for GPP, consistentwith strongly reduced model GPP and SIF at the time of the greenness maximum. We slightly reformulated these sentences and reordered this second paragraph of the discussion in order to make this point more clear. ###

P13, L24-34: a. Mean time lag is 6 to 11 days depending on SIF product. ++This was a misunderstanding due to unclear wording. The sentence has been reformulated such that it becomes clear that there is a time difference of 10 to 12 days between APAR proxies and SIF GFZ, depending on the APAR proxy. b. It would be helpful if the authors could plot a seasonal time series of APAR, GPP, and SIF with arrows indicating how changing light absorption affects the timing of peak GPP (lower light & temp) and SIF (higher light & temp), assuming the differential phasing is physical. ++We are afraid we are not sure we understand what is wanted here. The mean annual cycles of the vegetation proxies are shown in a new SI figure (as shown on the previous page) but without any arrows. c. The explanation of the difference between APAR and SIF is explained, but it's not clear from the discussion why instant SIF and SIF.cos(SZA) peak after GPP. The paragraph discusses the theory that light saturating conditions

leads to correlation of fluo- rescence and photosynthesis, suggesting the peaks are synchronized, not offset. Para- zoo et al. (2018) show synchrony of SIF, SIF driven GPP, and tower GPP in summer, but an offset in spring, suggesting an error in GPP data. However it's possible the SIF delay reflects conditions of diverging SIF and GPP under increased heat and water stress later in summer, with time of day again playing an important role in observed phasing. Thus, it is worth discussing all of the range of outcomes that the SIF delay is (a) physical, (b) related to error in model GPP as discussed later (P15, L22-34), and (c) negligible due to high uncertainty and variability in all datasets. ++We agree that this has not been fully addressed. We included in the discussion aspects of overpass time, cloud filter effects and a comparison to EC-derived in-situ GPP (new figure in SI, here Fig. 10 & 11) in order to get an indication of whether model GPP or SIF are closer to the in-situ patterns. This comparison, however, remains inconclusive. We used a subset of EC-sites from the FLUXNET 2015 data set for an additional comparison to satellite SIF, temperature (site-level, filled with down-scaled and regressed ERA-Interim) and FLUXCOM GPP (please note that FLUXCOM has been trained on the earlier LaThuile dataset). In addition, we were provided with EC data from the Cherski site in Russia by the PI (Mathias Göckede). As you say, site-level comparisons might be useful in order to get indications of whether SIF or model GPP are actually closer to site level GPP and whether there is indeed a relationship to temperature or not. We have summarized the time series as differences of the peak timing of the individual proxies to the annual temperature maximum per site and across sites in the plots below. We include them together as additional figures in the SI and discuss them. The results remain non conclusive. Model GPP and SIF have similar time lags to the temperature peak per site, but there is no consistent ordering of one before other for all sites. Importantly, site-GPP has overall similar time lags to temperature and also does not show consistency regarding the sign of the lags, not even between partitionings. So neither site-GPP, nor model GPP nor SIF shows consistent temporal behaviour with respect to temperature at site-level. Summarizing over all sites the time difference to temperature is actually largest for the tower GPP

(indicated as squares excluding the sites No-Adv, DK-ZaH, DK-ZaF where SIF shows only noise and either only one year of good NEE is available or the partitionings show very different behaviour, the mean across sites and years including all sites is shown as diamonds). Such comparisons are hampered by a number of issues: 1) scale-mismatch and spatial representativeness, 2) temporal overlap of site-data with satellite data, 3) data quality of NEE and temperature at site-level, 4) quality of partitioning at site-level (partly strongly different behaviour between partitioning methods, though they mostly converge versus the peak growing season). Please find attached plots for two of the 12 sites containing all the time series and quality information. These site-level time series illustrate these issues very well and clear suggest that EC cannot always be used as the 'truth' in evaluating satellite observations. These results strongly underline the observational problems in tundra outlined in the introduction of the manuscript and call for more in-situ measurements that are well characterized and understood in order to interpret the signals seen from the satellite. ###

Minor Comments P9, L27-29: 2009 typo? Somewhere in this line the authors appear to extrapolate from 2012 to 2009, but it's not specifically mentioned ++We clarified in the manuscript that 'We take 2009 as representative for the period of investigation. Based on information on the global tree cover in 2000, the yearly losses until 2009 and the gains until 2009 (assuming the growth between 2000 and 2012 indicated in the data set is linear), global tree cover in 2009 is estimated.' ###

P12, L11: "less frequently" is vague. Please quantify. ++Based on the new Fig. 7 we reformulate: 'The important role of energy-related variables, mostly temperature, for vegetation activity and growth is highlighted by widespread highest correlations with temperature and radiation for RSGPP, SIF GFZ, EVI and NDVI, while moisture-related variables are most important only in about one third of the pixels. VOD and NDVI.Rg do show a strong relationship with the annual temperature maximum only in 25 to 30% of the pixels and about half of the pixels show a higher importance of soil moisture or open water on the surface. ' ###

P12, L20: "shrubs and trees" -> more precisely, "mixed shrub-tree land covers" ++We changed it accordingly

P16, L4: "second third of July" is awkward. Might rephrase as "one week later" ++We corrected this. P16, L12: might check out parallel study by Parazoo et al (2018) ++Thanks for this reference, we do. Interactive comment on Biogeosciences Discuss., https://doi.org/10.5194/bg-2018-196, 2018.

[Figure]

**Fig. 1.**

[Figure]

**Fig. 2.**

[Figure]

**Fig. 3.** Mean seasonal cycles for the area at Laptev Strait indicated in Fig.1 of the manuscript. The values have been scaled to to 0-1 using the min/max range.

[Figure]

**Fig. 4.** Mean seasonal cycles for the area in the Siberian Forest indicated in Fig.1 of the manuscript. The values have been scaled to to 0-1 using the min/max range.

Maximum absolute full spearman rank correlation

[Figure]

**Fig. 5.** Frequency distribution of pixelwise spearman rank correlation between the peak DOY of the vegetation proxies across years and the peak DOY of environmental variables in spatial moving windows of 1.5\

[Figure]

**Fig. 6.** Mean seasonal cycle

[Figure]

**Fig. 7.** Mean seasonal cycle

[Figure]

**Fig. 8.** Mean seasonal cycle

[Figure]

**Fig. 9.** Mean seasonal cycle

[Figure]

**Fig. 10.** Difference in peak timing of the different indicators to site-level temperature over years at selected FLUXNET sites. Diamonds represent the average peak lag across all years per site and variable.

**Fig. 11.** Difference in peak timing of the different indicators to site-level temperature over years and at selected FLUXNET sites. Diamonds represent the average peak lag across all years and sites, while squa

[Figure]

[Figure]

[Figure]

[Figure]

[Figure]

**Fig. 12.** RU-Sam (Samoylov):Gray bars: fractions of good measured and gap-filled NEE data, blue bars: the same for temperature data

[Figure]

[Figure]

[Figure]

[Figure]

[Figure]

**Fig. 13.** DK-NuF (Greenland):Gray bars: fractions of good measured and gap-filled NEE data,
blue bars: the same for temperature data

---

## Author Comment (AC2) · 10 Jul 2018

Summary: This paper analyzes the timing of peak vegetation productivity for high latitude treeless ecosystems based on spaceborne remote sensing observations. Satellite data utilized in the study includes: 1) MODIS-based GPP; 2) MODIS NDVI; 2) MODIS-based APAR; 3) GOME2 SIF; and 4) AMSR VOD. The authors find a consistent ordering of peaks: APAR < GPP < SIF < VIs / VOD. The authors conclude that the consistent differences between photosynthetic activity and greenness is an important consideration when using satellite observations as drivers in vegetation models. Overall, I find this to be a nice paper with some interesting/useful findings. I do however have a number of recommended revisions before I can recommend this paper for publication. Most importantly, I highlight a number of ways to potentially improve the analysis and better demonstrate robustness of the findings.

Major Comments: 1. The introduction is very well written and does an excellent job of framing the research questions and establishing the importance of the work. 2. The processing of the various satellite observations is explained very well and done appropriately.

++Thank you. ###

3. The authors use a definition of the timing of peak vegetation productivity as "the timing of the annual maximum is defined as the average day of year (DOY) of all days at which the values exceed the 95th quantile of all valid values of the time series in a year in a given pixel." I question this method as these different data have inherently different levels of daily variability, which could result in spurious dates being included in the period exceeding the 95th quantile of all valid values. I recommend filtering (e.g., using a Savitzky-Golay filter) and then finding the peak of the time series. This would be a good check on the robustness of the main findings of the paper.

++We fully agree that different levels of data availability and variability renders the results sensitive to the method of peak identification and confirm that spurious effects of large values especially in the beginning and at the end of the growing season (looking at random pixel time series, not shown) affect the peak timing (examples e.g. in the manuscript in Figures A5 and A7 , or as also shown for SIF GFZ in the time series at site level in the plots for the EC site-level evaluation below) . These effects are stronger for some vegetation proxies than for others. We had also tested other methods for smoothing and the peak identification (including function fitting as in Gonsamo et al. 2013 Ecological Indicators), but did not find one method that could accommodate all problems and therefore decided to stay with the most straightforward method. Despite large spread and only a limited amount of years in the analysis we think that the ordering of the groups of proxies in Fig.4 supports robustness on the large scale. In order to still test for robustness and obtain uncertainties we bootstrapped the time series (time series per proxy and year are resampled each 50 times consistently across proxies, i.e. the same DOYs are sampled for each proxy) and calculated the peak time for the bootstrap samples with the same method (average of all DOYs exceeding the 95th percentile). Preliminary results for selected proxies are shown below in fig.1: ###

4.Additionally, I find the examination of the peak of seasonal activity interesting, but feel it would be complemented by consideration of the start and end of season. Using the above-mentioned Savitzky-Golay filter (or something similar), start and end of season estimates could be easily derived. I recognize this is a lot of additional work and analysis, but I think it could really strengthen the findings of the paper. I put this forward as a recommendation that could strengthen the paper, but this additional analysis is not necessary needed to warrant publication as the current findings of the paper are still very interesting.

++Indeed, analysis of the complete growing season would make the study more complete. However, in tundra ecosystems the onset and end of the growing season are very fast transitions and considering the difficult observation conditions and high noise levels, accurate characterization of the sharp SOS and EOS will be more challenging. We therefore decided to focus here on the peak growing season. ###

5. I think the study would benefit from comparison at any eddy co- variance flux tower sites within the study area. EC flux tower-derived GPP data could be used more effectively than modeled GPP to establish which proxies are capturing peak photosynthesis and which are capturing other aspects of plant dynamics (e.g., changes in water content, changes in leaf area, etc.). In particular, this would be useful in establishing whether SIF observations are providing new information more directly associated with plant physiological function. For instance, the authors may find SIF bet- ter matches EC tower-based GPP compared to the modeled GPP used in the analysis. This would be very useful information for the modeling community.

++We used a subset of EC-sites from the FLUXNET 2015 data set for an additional comparison to satellite SIF, temperature (site-level, filled with down-scaled and regressed ERA-Interim) and FLUXCOM GPP (please note that FLUXCOM has been trained on the earlier LaThuile dataset). In addition, we were provided with EC data from the Cherski site in Russia by the PI (Mathias Göckede). As you say, site-level comparisons might be useful in order to get indications of whether SIF or model GPP are actually closer to site level GPP and whether there is indeed a relationship to temperature or not. We have summarized the time series as differences of the peak timing of the individual proxies to the annual temperature maximum per site and across sites in the plots below (Fig.2&3). We include them together as additional figures in the SI and discuss them. The results remain non conclusive. Model GPP and SIF have similar time lags to the temperature peak per site, but there is no consistent ordering of one before other for all sites. Importantly, site-GPP has overall similar time lags to temperature and also does not show consistency regarding the sign of the lags, not even between partitionings. So neither site-GPP, nor model GPP nor SIF shows consistent temporal behaviour with respect to temperature at site-level. Summarizing over all sites the time difference to temperature is actually largest for the tower GPP (indicated as squares excluding the sites No-Adv, DK-ZaH, DK-ZaF where SIF shows only noise and either only one year of good NEE is available or the partitionings show very different behaviour, the mean across sites and years including all sites is shown as diamonds). Such comparisons are hampered by a number of issues: 1) scale-mismatch and spatial representativeness, 2) temporal overlap of site-data with satellite data, 3) data quality of NEE and temperature at site-level, 4) quality of partitioning at site-level (partly strongly different behaviour between partitioning methods, though they mostly converge versus the peak growing season). Please find attached plots for two of the 12 sites (Fig. 4&5) containing all the time series and quality information. These site-level time series illustrate these issues very well and clearly suggest that EC cannot always be used as the 'truth' in evaluating satellite observations. These results strongly underline the observational problems in tundra outlined in the introduction of the manuscript

and call for more in-situ measurements that are well characterized and understood in order to interpret the signals seen from the satellite. ###

6. Page 16, line 15: "Furthermore, the fact that the SIF maximum is reached in close temporal agreement with air temperature indicates a benefit for photosynthesis from highest temperatures." Without comparing to actual GPP observations (e.g., from EC flux towers), I do not think this is a valid conclusion. There is nothing in this analysis to conclusively show SIF is actually tracking GPP since it is only compared to modeled GPP.

+++Yes, this cannot conclusively be shown, especially considering the above results at site-level. Still, both model GPP and SIF peak in closest temporal agreement with temperature compared to APAR-proxies and greenness at the satellite scale. We therefore underlined the hypothetical nature of this sentence by changing it to: "Furthermore, the fact that the SIF maximum is reached in close temporal agreement with air temperature might indicate a benefit for photosynthesis from highest temperatures." ###

Minor Comments: Page 7, line 15: time should be corrected to "1:30 AM". Page 8, Line 14: grammar error: "and it using such scaling factor may further amplify noise." ++Both have been corrected in the manuscript, thanks for this. Interactive comment on Biogeosciences Discuss., https://doi.org/10.5194/bg-2018-196, 2018.
* * *
[Figure]

**Fig. 1.** Lag between the peak timing of the different proxies and the peak doy of NDVI based on 50 resampled time series per proxies, pixel, year. Shown are only the significant (95%) differences.

[Figure]

**Fig. 2.** Difference in peak timing of the different indicators to site-level temperature over years at selected FLUXNET sites. Diamonds represent the average peak lag across all years per site and variable.

[Figure]

**Fig. 3.** Difference in peak timing of the different indicators to site-level temperature over years and at selected FLUXNET sites. Diamonds represent the average peak lag across all years and sites, while squa

[Figure]

[Figure]

[Figure]

[Figure]

[Figure]

**Fig. 4.** DK-NuF: Gray bars: fractions of good measured and gap-filled NEE data, blue bars: the same for temperature data

[Figure]

[Figure]

[Figure]

[Figure]

[Figure]

**Fig. 5.** RU-Sam: Gray bars: fractions of good measured and gap-filled NEE data, blue bars: the same for temperature data